# MAS-Architect: Declarative Multi-Agent System Design via Separation of Concerns

Jing Huang [* 1]  Lidong Zhang [* 1]  Mutian Bao [* 1]  Yadong Li [2]  Xingzhong Xu [2]  Jinjian Zhang [2]  Jie Liu [3]
Ming Kong [1]  Qiang Zhu [1]

## Abstract

The Automated Design of Multi-Agent Systems (Auto-MAS) has emerged as a promising framework for addressing complex reasoning tasks. However, existing approaches often suffer from structural rigidity and entangle the design of system topology with the implementation of individual agents. To overcome these limitations, we propose MAS-Architect, a framework that automates MAS design through a novel code-based declarative MAS paradigm rooted in the *Separation of Concerns* principle. By decoupling topology planning from node implementation via a unified interface, our approach enables the from-scratch generation of task-adaptive architectures. We further employ a *Distill-then-Explore* training strategy to optimize these designs. Comprehensive experiments on five benchmarks show that MAS-Architect sets a new Pareto frontier in the efficiency–performance trade-off: it surpasses state-of-the-art methods while substantially lowering token usage. Notably, the framework achieves a strong average accuracy of 78.7% across benchmarks with an inference cost of only 2,533 tokens per query. Qualitative analysis reveals the autonomous emergence of advanced collaboration patterns, validating the generative flexibility of the declarative paradigm. Code will be available at https://github.com/ZJUHJ/mas_architect.

## 1. Introduction

Large Language Model (LLM)-driven Multi-Agent Systems (MAS) have emerged as the dominant paradigm for solving complex tasks (Chen et al., 2024). By leveraging role specialization and collaborative interaction, MAS continue to push performance boundaries across domains such as mathematical reasoning, code generation, and general-purpose QA (Hong et al., 2023; Qian et al., 2024; Chen et al., 2025; Jin et al., 2025). Despite these advances, constructing high-performing MAS remains labor-intensive, as it requires careful manual tuning of agent roles, interaction structures, and communication protocols through extensive expert-driven iteration. To mitigate this burden, Automated Design of MAS (Auto-MAS) has become a focal point of research (Hu et al., 2024).

Existing Auto-MAS methods primarily adopt two representation paradigms, yet both exhibit notable limitations, as illustrated in Figure 1. The **Graph-based paradigm** instantiates MAS as a Directed Acyclic Graph (DAG) (Zhang et al., 2024a; Zhuge et al., 2024; Li et al., 2025; Zhang et al., 2025), offering formal rigor but suffering from inherent rigidity. This approach restricts node generation by compelling agents to be selected from predefined role pools or operator libraries, which limits the dynamic creation of customized nodes tailored to specific task needs. Such inflexibility extends to topology exploration, where the reliance on pruning or modifying fixed template graphs hinders the construction of novel collaboration structures from scratch. Moreover, the static routing nature of DAGs inherently limits control flow expressiveness, rendering it difficult to capture complex, adaptive patterns such as conditional branching and loop iterations. Conversely, recent works treat Auto-MAS as a code generation problem, using a Meta-Agent to represent MAS as executable programs (Hu et al., 2024; Zhang et al., 2024b; Ye et al., 2025). While achieving Turing-complete expressiveness, most current representative implementations adopt an **Imperative Paradigm**, which tightly couples collaboration topology, control flow logic, and node implementation within a single procedural code stream. This coupling causes the topological structure to be implicitly embedded and difficult to optimize independently,

---

*Equal contribution [1]Zhejiang University [2]Ant Group [3]City University of Hong Kong. Correspondence to: Yadong Li <liyadong.lyd@antgroup.com>, Ming Kong <zjukong-ming@zju.edu.cn>, Qiang Zhu <zhuq@zju.edu.cn>.

*Proceedings of the 43rd International Conference on Machine Learning*, Seoul, South Korea. PMLR 306, 2026. Copyright 2026 by the author(s).

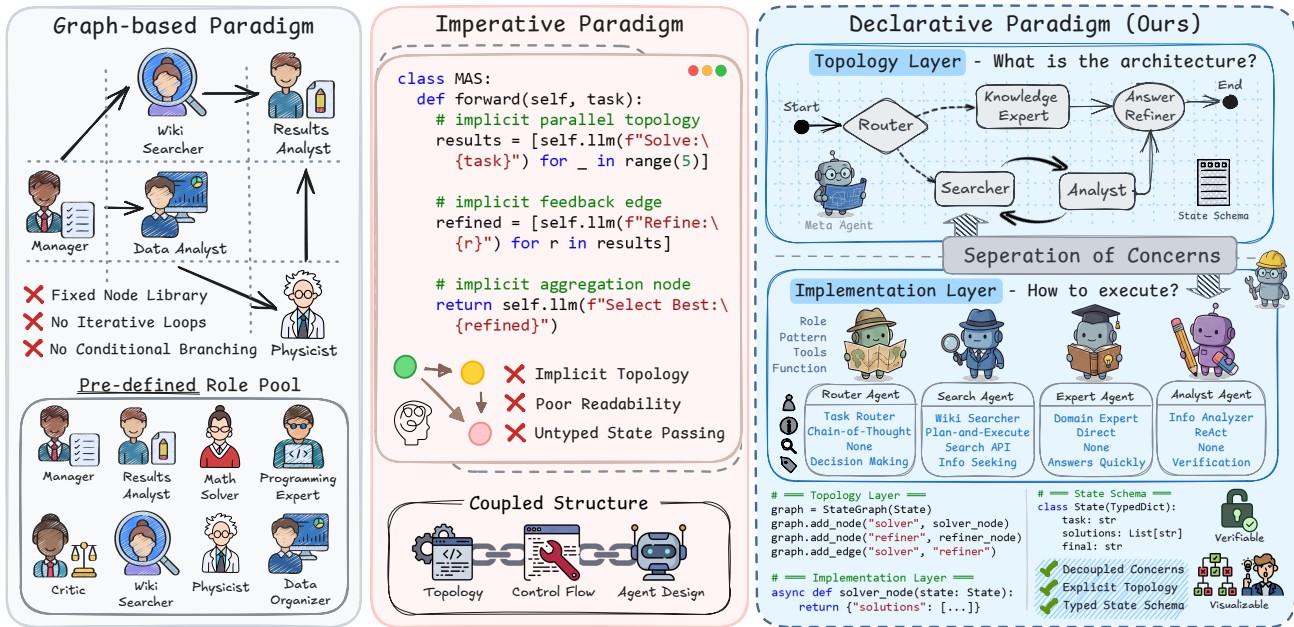

Figure 1. Comparison of MAS representation paradigms.

while the absence of explicit typed semantics for state passing further degrades code readability and maintainability.

To address the complementary deficiencies of these two paradigms, we propose the **Declarative MAS Paradigm**, a novel code-based representation for Auto-MAS. As shown in Figure 1, inspired by the *Separation of Concerns* (Dijkstra, 1982) principle in software engineering, our core idea is to decompose MAS design into two orthogonal abstraction layers: (1) The **Topology Layer** employs a declarative graph definition—which is independently searchable, visible, and formally verifiable—to explicitly declare dynamic control flows (e.g., branching, loops), describing *"what the architecture is"*; (2) The **Implementation Layer** is responsible for the specific realization of each agent node, focusing on *"how to execute."* These two layers are decoupled via a standardized **State Schema** interface, which autonomously maintains dynamic context (e.g., task information, intermediate results, and interaction trajectories). This layered design combines the structural clarity of graph methods with the expressive completeness of code methods. Crucially, this structured decomposition provides a viable path for **Task-Adaptive From-Scratch Generation**: the Meta-Agent no longer relies on predefined templates or role libraries, but instead independently searches for collaboration structures at the topology layer and synthesizes node specifications on demand at the implementation layer, coordinating both via the state schema.

Based on the Declarative MAS Paradigm, we propose **MAS-Architect**, an end-to-end framework for the automatic generation of customized MAS from task queries. MAS-Architect

fully leverages the decoupled nature of the declarative language to stratify the generation process into topological planning and node implementation. At the topology layer, the Meta-Agent infers the necessary collaboration patterns based on task semantics, autonomously determining node count, connection relationships, conditional routing rules, and state schema definitions to build a task-adapted topology from scratch. At the implementation layer, for each node in the topology, the Meta-Agent synthesizes an agent's complete execution specification (Luo et al., 2025)—including profiling (e.g., Manager, Critic, Executor), planning patterns (e.g., CoT (Wei et al., 2022), ReAct (Yao et al., 2022), Reflexion (Shinn et al., 2023)), and tool bindings. This stratified generation strategy not only reduces the complexity of single-pass generation but also enables independent iterative optimization of topology and node implementation.

For training, we adopt a two-stage *Distill-then-Explore* paradigm. (1) **Architecture Distillation**: We utilize a large teacher model to generate declarative MAS specifications for diverse queries. After filtering out execution failures or incorrect answers via rejection sampling, we construct a high-quality corpus to enable a smaller Meta-Agent to rapidly acquire reasonable architectural design patterns via supervised fine-tuning. (2) **Architecture Exploration**: Initialized with the distilled policy, the model undergoes policy exploration via Reinforcement Learning with Verified Rewards (RLVR). Because the search space of the declarative MAS is well-structured—policy updates have clear semantics at the topology level—RL exploration is efficient and stable, guiding the model to emerge with novel collaboration

structures that surpass the teacher's priors. We posit that the key to successfully acquiring dynamic architecture design capabilities lies in the *structural inductive bias* provided by the declarative representation: whereas imperative code mixes topology, control flow, and implementation logic, forcing the model to implicitly infer structure from verbose procedural code (resulting in sparse and ambiguous supervision signals), the declarative paradigm explicitly symbolizes the topology. This allows the model to directly learn the structured mapping from task requirements to collaboration topology, significantly reducing learning complexity.

We systematically evaluated our approach on benchmarks, including mathematical reasoning, general reasoning, and multi-hop QA. Experiments demonstrate that MAS-Architect outperforms the existing SOTA method MAS-GPT (Ye et al., 2025) by 2.9% while reducing token consumption by 23.8% on GSM8K (Cobbe et al., 2021), establishing a new efficiency-performance Pareto frontier. Furthermore, we observed the autonomous emergence of novel topological patterns (e.g., parallelized reasoning-exploration streams, hierarchical dual-loop refinement), validating the generative flexibility of the declarative paradigm.

Our contributions are as follows:

- We propose the **Declarative MAS Paradigm**, leveraging the principle of Separation of Concerns to overcome the expressive limitations of graph methods and the coupling dilemmas of imperative code generation. This orthogonal decomposition of architecture design and node implementation provides a new theoretical foundation and design space for the field.

- We introduce **MAS-Architect**, an end-to-end framework that abandons reliance on predefined role libraries and template graphs. Through a task-adaptive from-scratch generation mechanism and a *Distill-then-Explore* training strategy, it empowers smaller models with the capability to create query-level customized MAS.

- We conduct comprehensive experiments on five benchmarks, quantitatively validating the advantages of MAS-Architect in performance, efficiency, and generalization, and revealing the architectural design principles acquired by the model through the analysis of emergent structures.

## 2. Related Work

### 2.1. Interaction Patterns of Large Language Models

The application paradigm of Large Language Models (LLMs) has undergone a profound evolution, transitioning from simple responsive interactions to complex autonomous agent systems. Early research primarily focused on instruction tuning to enable models to follow human instructions for basic question-answering tasks (Ouyang et al., 2022). However, direct input-output mappings proved insufficient for complex logical reasoning. To address this, Wei et al. introduced Chain-of-Thought (CoT) prompting (Wei et al., 2022), which significantly enhances performance by eliciting intermediate reasoning steps. Building on this, Wang et al. proposed Self-Consistency (Wang et al., 2022) to further optimize the reasoning process. By sampling diverse reasoning paths and employing majority voting, this approach effectively mitigates the uncertainty inherent in single-chain reasoning and improves robustness. To overcome the limitations of relying solely on internal knowledge, Yao et al. introduced the ReAct framework (Yao et al., 2022), which innovatively synergizes reasoning and acting to equip models with the capability to utilize external tools. Subsequently, interaction forms have evolved toward multi-perspective collaboration. For instance, the LLM-Debate mechanism proposed by Du et al. (Du et al., 2023) demonstrates that multi-agent debate and iterative interaction can effectively rectify misconceptions and enhance factual consistency.

### 2.2. Automated Design of Multi-Agent Systems

The reliance of Multi-Agent Systems (MAS) on intricate, hand-crafted topologies has proven to be a bottleneck for scalability, prompting a shift toward automated design. Initial efforts, such as GPTSwarm (Zhuge et al., 2024) and AFlow (Zhang et al., 2024b), adopted task-level iterative optimization via reinforcement learning or Monte Carlo Tree Search. However, these methods produce a one-size-fits-all architecture for a dataset rather than customizing the MAS for each specific query. This limitation led to the development of query-level frameworks like G-Designer (Zhang et al., 2024a), ADAS (Hu et al., 2024), ARG-Designer (Li et al., 2025), and MaAS (Zhang et al., 2025), which leverage autoencoders or supernet sampling for adaptive construction. Despite this progress, such methods are often confined by predefined pools of agent roles or operators, restricting their adaptability to new functions. Addressing this, MAS-GPT (Ye et al., 2025) proposed the "MAS-as-Code" paradigm, treating system construction as a Python-based generative language task. This allows for the dynamic generation of prompts and logic in a single inference pass, bypassing predefined constraints. Nonetheless, the flexibility of this paradigm is currently bounded by the topological variations and logical patterns encompassed in the training dataset.

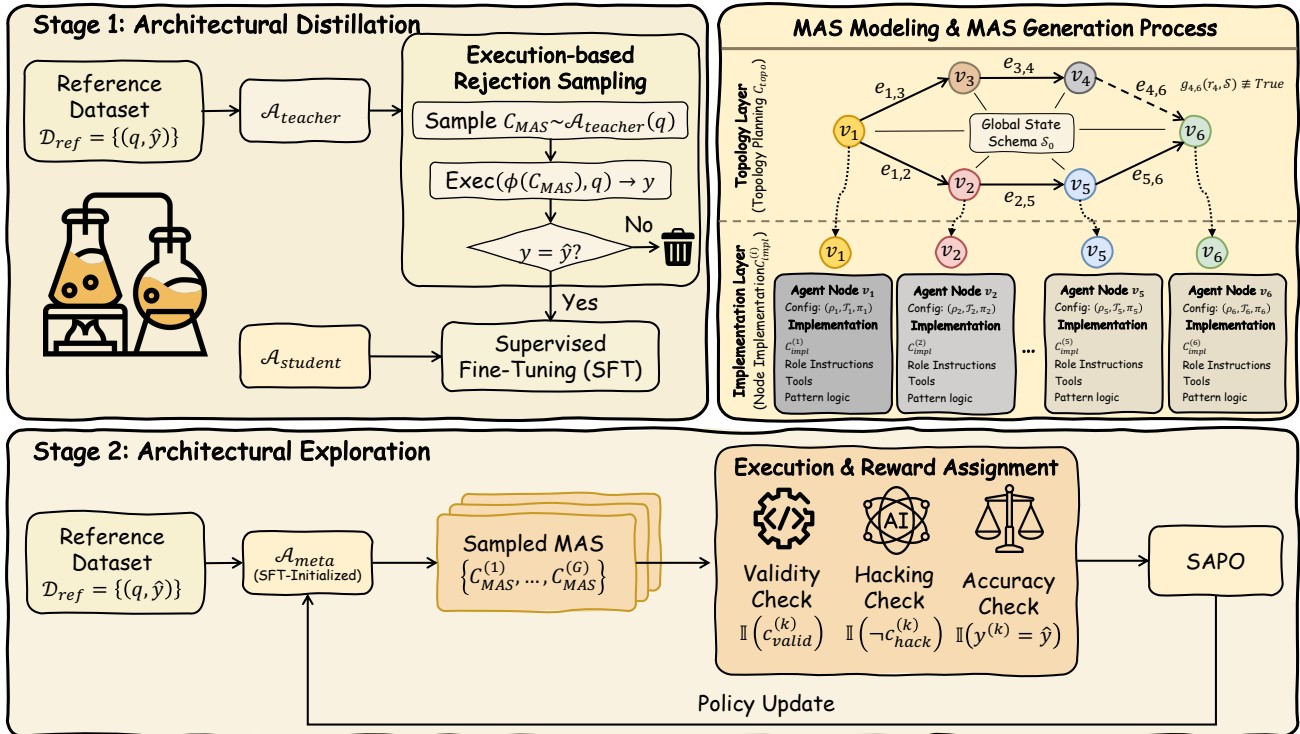

*Figure 2.* The MAS-Architect framework. The system synthesizes query-specific MAS via Topology Planning and Node Implementation (right). The training pipeline employs a *Distill-then-Explore* strategy (left), progressing from execution-based Supervised Fine-Tuning (Stage 1) to Reinforcement Learning (Stage 2).

## 3. Methodology

### 3.1. Formalization

**Task Definition.** We formulate the task of generating a response $y$ for a query $q \in \mathcal{Q}$ as the dynamic construction and execution of a Multi-Agent System (MAS). In the *Construction Phase*, the Meta-Agent $\mathcal{A}_{meta}$ synthesizes a query-specific MAS, denoted as $\mathcal{G}$:

$$\mathcal{G} = \mathcal{A}_{meta}(q). \tag{1}$$

Subsequently, in the *Execution Phase*, the system executes the constructed MAS to derive the final response:

$$y = \text{Exec}(\mathcal{G}, q). \tag{2}$$

In the following, we provide the formal definition of the MAS $\mathcal{G}$ and its constituents.

**MAS Modeling.** We formalize the Multi-Agent System (MAS) as a stateful directed computation graph $\mathcal{G} = (V, E, \mathcal{S}_0)$. Here, $\mathcal{S}_0$ denotes the initial *Global State* schema, which instantiates the runtime state $\mathcal{S}$.

The vertex set $V$ consists of agent nodes. Each agent $v_i \in V$ is defined by the tuple $v_i = \langle \rho_i, \mathcal{T}_i, \pi_i, \mathcal{H}_i \rangle$, where $\rho_i$ (*role*), $\mathcal{T}_i$ (*tools*) and $\pi_i$ (*patterns* of reasoning, e.g., CoT, ReAct,

etc.) are static meta-attributes, while $\mathcal{H}_i$ denotes the dynamic *Local State*. Prior to activation, $\mathcal{H}_i$ is initialized with the injected task context; during execution, it is incrementally updated to record the agent's reasoning trajectory. Formally, an activated agent $v_i$ executes a mapping function $f_i : (\rho_i, \mathcal{T}_i, \pi_i, \mathcal{H}_i, \mathcal{S}) \rightarrow r_i$. The resulting action $r_i$ is merged into the global state via the update operator $\mathcal{S} \leftarrow \mathcal{S} \oplus r_i$, where $\oplus$ denotes the incremental update of the MAS state based on the current agent node's execution result.

The edge set $E$ defines the interaction topology. Each edge $e_{i,j} \in E$ is governed by a *Transition Protocol* containing a guard function $g_{i,j}(\cdot)$. A control flow transition from $v_i$ to $v_j$ occurs if and only if the condition $g_{i,j}(r_i, \mathcal{S}) = \text{True}$ is satisfied. This formalism unifies both deterministic connections (where $g_{i,j} \equiv \text{True}$) and conditional routing, allowing the computation path to adapt dynamically to the execution state.

### 3.2. MAS-Architect

Figure 2 shows MAS-Architect, which generates the MAS via declarative code construction, decomposing the process into two phases: *Topology Planning* synthesizes the architecture, while *Node Implementation* concretizes agent logic.

**Topology Planning.** The primary objective of this phase is to produce a declarative MAS specification of the MAS $\mathcal{G}$. $\mathcal{A}_{meta}$ first analyzes the query $q$ to define the initial *Global State* schema $\mathcal{S}_0$, which serves as the typed interface for inter-agent communication and state management.

Subsequently, $\mathcal{A}_{meta}$ constructs the topology by synthesizing the structural definition code $C_{\text{topo}}$:

$$C_{\text{topo}} = \mathcal{A}_{meta}^{\text{topo}}(q). \tag{3}$$

The execution of $C_{\text{topo}}$ instantiates the vertex set $V$, the edge set $E$, and the global schema $\mathcal{S}_0$. Crucially, for each agent node $v_i \in V$, the Meta-Agent determines its static configuration tuple $\langle \rho_i, \mathcal{T}_i, \pi_i \rangle$, thereby establishing the functional blueprint of the agent. Regarding the interaction topology, for every conditional edge $e_{i,j} \in E$, $\mathcal{A}_{meta}^{\text{topo}}$ generates the guard function $g_{i,j}(r_i, \mathcal{S})$. This function encapsulates the control logic by evaluating the predecessor's output $r_i$ and the current global state $\mathcal{S}$ to determine the validity of the transition. For the specific implementation of $C_{\text{topo}}$, please refer to Figures 13 and 15.

**Node Implementation.** With the topology and node roles established, the process transitions to instantiating the functional kernels for each agent node $v_i \in V$. While the previous stage defined the static attributes $\langle \rho_i, \mathcal{T}_i, \pi_i \rangle$, this stage synthesizes the executable logic governing the mapping function $f_i$. The Meta-Agent generates the specific implementation code $C_{\text{impl}}^{(i)}$ for each node, conditioned on its configuration and the query:

$$C_{\text{impl}}^{(i)} = \mathcal{A}_{meta}^{\text{impl}}(\langle \rho_i, \mathcal{T}_i, \pi_i \rangle \mid q). \tag{4}$$

The generated code $C_{\text{impl}}^{(i)}$ encapsulates the concrete execution details by strictly adhering to the node's configuration. Specifically, it constructs system instructions aligned with the role $\rho_i$, binds the executable interfaces for tools $\mathcal{T}_i$, and operationalizes the reasoning pattern $\pi_i$ to derive the action $r_i$ and trigger the global state update $\mathcal{S} \leftarrow \mathcal{S} \oplus r_i$. The final executable specification is obtained by assembling the topology definitions and node implementations:

$$C_{\text{MAS}} \triangleq \text{Assemble}\left( C_{\text{topo}}, \{C_{\text{impl}}^{(i)}\}_{i=1}^{|V|} \right). \tag{5}$$

Compiling this specification instantiates the logical graph $\mathcal{G} = \phi(C_{\text{MAS}})$, where $\phi$ is the compilation operator. For the specific implementation of $C_{\text{impl}}^{(i)}$, please refer to Figures 12 and 14.

### 3.3. Training Strategy: Distill-then-Explore

We employ a two-stage training paradigm to equip the MAS-Architect with the capability to autonomously design query-level MAS: Stage-1: Architectural Distillation injects architectural thinking by learning from expert corpora, while Stage-2: Architectural Exploration transcends imitation through result-oriented self-exploration to foster the emergence of diverse, effective MAS patterns for query resolution.

**Architectural Distillation.** The objective of this phase is to distill the architectural design capabilities of a superior teacher model $\mathcal{A}_{\text{teacher}}$ into the Meta-Agent. We start with a reference dataset $\mathcal{D}_{\text{ref}} = \{(q, \hat{y})\}$. For each query, we sample a candidate specification $C_{\text{MAS}} \sim \mathcal{A}_{\text{teacher}}(q)$. This candidate is then compiled into a graph $\mathcal{G}$ and executed to derive a response $y = \text{Exec}(\mathcal{G}, q)$. To ensure data quality, we employ *Execution-based Rejection Sampling*, retaining only the instances where the execution result $y$ is consistent with the ground truth $\hat{y}$. This yields the final instruction tuning corpus:

$$\mathcal{D}_{\text{SFT}} = \Big\{(q, C_{\text{MAS}}) \,\Big|\, (q, \hat{y}) \in \mathcal{D}_{\text{ref}}, \ C_{\text{MAS}} \sim \mathcal{A}_{\text{teacher}}(q),$$
$$y = \text{Exec}(\phi(C_{\text{MAS}}), q), \ y = \hat{y}\Big\}. \tag{6}$$

Subsequently, we perform Supervised Fine-Tuning (SFT) on the Meta-Agent to minimize the autoregressive loss on the validated code samples:

$$\mathcal{L}_{\text{SFT}}(\theta) = -\mathbb{E}_{(q, C_{\text{MAS}}) \sim \mathcal{D}_{\text{SFT}}} \left[ \sum_{t=1}^{|C_{\text{MAS}}|} \log P_\theta(z_t \mid q, z_{<t}) \right]. \tag{7}$$

**Architectural Exploration.** To transcend the limitations of the teacher's distribution and encourage the emergence of novel topologies, we transition to Reinforcement Learning (RL) using the SAPO algorithm (Gao et al., 2025). We initialize the policy $\mu_\theta$ of the Meta-Agent $\mathcal{A}_{meta}$ with the SFT model. During this phase, for each query $q$, the policy generates a group of $G$ candidate specifications $\{C_{\text{MAS}}^{(k)}\}_{k=1}^G \sim \mu_\theta(\cdot|q)$. Each candidate is compiled into a graph $\mathcal{G}^{(k)}$ and executed to derive a response $y^{(k)}$. To guide the optimization, we design a hierarchical reward function $R(C_{\text{MAS}}^{(k)}, q)$ that strictly enforces validity and honesty. Specifically, we employ an LLM-based auditor to detect "shortcut learning" (e.g., hard-coding answers), yielding a boolean flag $c_{\text{hack}}^{(k)}$. The total reward is formulated as:

$$R(C_{\text{MAS}}^{(k)}, q) = \mathbb{I}(\mathcal{G}_{\text{valid}}^{(k)}) \cdot \mathbb{I}(\neg c_{\text{hack}}^{(k)})$$
$$\cdot \left( 0.1 + 0.9 \cdot \mathbb{I}(y^{(k)} = \hat{y}) \right), \tag{8}$$

where $\mathbb{I}(\mathcal{G}_{\text{valid}}^{(k)})$ indicates successful compilation and execution. Under this formulation, invalid or cheating solutions satisfy the condition $R = 0$, while valid but incorrect attempts receive a base reward of $0.1$, reaching $1.0$ only upon

correct reasoning. Finally, the policy $\mu_\theta$ is updated to maximize the expected reward.

## 4. Experiments

### 4.1. Experimental Setup

**Benchmarks.** To comprehensively verify the generalizability of the proposed method, we evaluate it on 5 benchmarks across 2 domains. Specifically, we assess performance on GSM8K (Cobbe et al., 2021) (test-split), GSM-Hard (Gao et al., 2022), and MATH (Hendrycks et al., 2021) (test-split) for mathematical reasoning; MMLU (Hendrycks et al., 2020) (test-split) and GPQA (Rein et al., 2024) (Diamond) for general reasoning. In addition, we conducted ablation studies on Hotpot-QA (Yang et al., 2018) (val-split) in the domain of multi-hop reasoning.

**Metrics.** We follow the evaluation protocol of MAS-GPT (Ye et al., 2025). For queries with ground truth, we utilize Qwen3-30B-A3B-Instruct-2507 (Team, 2025) to extract the MAS execution results, compare it against the ground truth, and report the accuracy. For computational cost, we follow the G-Designer (Zhang et al., 2024a) setup and measure the prompt token consumption incurred by each MAS during execution.

**Baselines.** We compare the proposed method against methods chosen to represent three orthogonal paradigms: (1) single-agent baselines, such as Vanilla and CoT (Wei et al., 2022), Self-Refine (Madaan et al., 2023); (2) graph-based Auto-MAS approaches like GPTSwarm (Zhuge et al., 2024) representing optimization over fixed role pools; and (3) code-based Auto-MAS methods, including MAS-GPT (Ye et al., 2025) and DyLAN (Liu et al., 2023), which exemplify imperative code generation strategies.

**Implementation Details.** We employ Qwen3-Coder-30B-A3B-Instruct (Team, 2025) as the policy model to generate the MAS, and unless otherwise specified, Qwen3-4B-Instruct-2507 (Team, 2025) is used to drive the MAS. The training phase comprises Architectural Distillation, for which we sampled 28231 MMLU and 14760 HotpotQA pairs, and Architectural Exploration, utilizing 7405 HotpotQA, 6394 MMLU, and 7473 GSM8K pairs. We utilize Qwen3-Coder-480B-A35B-Instruct (Team, 2025) as the teacher model for distillation and Qwen3-30B-A3B-Instruct-2507 (Team, 2025) as the judge for exploration; all experiments were conducted on 8 × Nvidia H20 96GB GPUs. Detailed training hyperparameters for both stages are provided in Table 5 and Table 6. The prompts used in model training can be found in Appendix F.

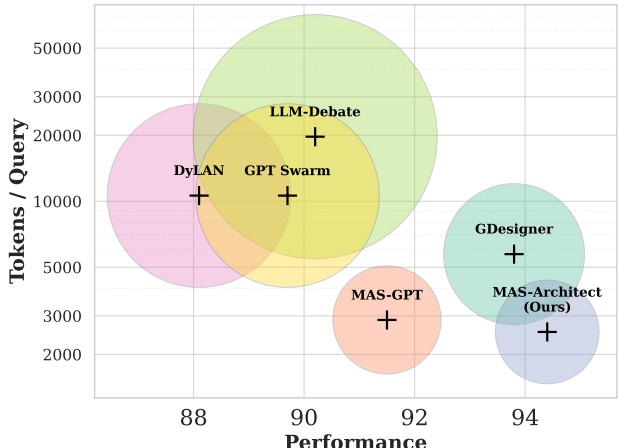

*Figure 3.* Efficiency Analysis on GSM8K. MAS-Architect (ours) achieves highest accuracy at lowest cost.

### 4.2. Main Results

Table 1 provides a comprehensive performance comparison across mathematical and general reasoning benchmarks, demonstrating that our proposed method, driven by the Qwen3-4B-Instruct-2507 model, consistently outperforms the vanilla baseline with substantial gains. Crucially, our approach demonstrates exceptional generalization capabilities on benchmarks explicitly excluded from the training data (marked with †), achieving significant accuracy improvements. Specifically, on the unseen MATH and GPQA benchmarks, our method reaches accuracies of 89.5 and 63.8, representing improvements of +7.4% and +8.7% over the vanilla baseline, respectively. Similarly, on the held-out GSM-Hard benchmark, we attain a competitive accuracy of 64.0, yielding a remarkable gain of +10.2%. Beyond these held-out tasks, the method secures leading accuracies of 94.4 on GSM8K (+3.6%) and 81.7 on MMLU (+1.9%). Consequently, our approach achieves an average accuracy of 78.7, surpassing the vanilla baseline by +6.4% and the second-best method by +3.0%.

### 4.3. Ablation Studies

As shown in Table 2, we conduct a comprehensive ablation study on the Architectural Distillation (w/ SFT) and Architectural Exploration (w/ RL) stages across varying model scales (Qwen3-4B-Instruct-2507 and Qwen3-30B-A3B-Instruct-2507). The results demonstrate a clear progressive enhancement pattern: while the initial SFT stage establishes a solid foundation with stable performance increments ranging from 1.0% to 2.4% across both model sizes, the subsequent RL stage acts as the primary catalyst for unlocking the models' latent potential. This impact is particularly pronounced in the 4B model, where the RL stage yields substantial margins of 7.4% on MATH, 9.0% on GPQA, and

*Table 1.* Performance comparison of different methods across multiple benchmarks. Note that Qwen3-4B denotes Qwen3-4B-Instruct-2507, and benchmarks marked with † are not included in the training data. The best results are **bolded** and the second-best results are underlined.

| METHOD | DRIVING MODEL | MATH | | | GENERAL REASONING | | AVERAGE |
|---|---|---|---|---|---|---|---|
| | | GSM8K | GSM-HARD† | MATH† | MMLU | GPQA† | |
| VANILLA | QWEN3-4B | 90.8 | 53.8 | 82.1 | 79.8 | 55.1 | 72.3 |
| VANILLA | GPT-4O-MINI | 87.5 | 58.0 | 78.2 | 78.6 | 38.0 | 68.1 |
| COT (WEI ET AL., 2022) | QWEN3-4B | 94.0 | **64.7** | 82.9 | 79.9 | 57.1 | 75.7 |
| LLM-DEBATE (DU ET AL., 2023) | GPT-4O-MINI | 89.5 | 60.8 | 79.6 | 80.8 | 37.8 | 69.7 |
| SELF-REFINE (MADAAN ET AL., 2023) | GPT-4O-MINI | 87.5 | 54.6 | 74.6 | 79.2 | 33.3 | 65.8 |
| AGENTVERSE (CHEN ET AL., 2023) | GPT-4O-MINI | 89.9 | 55.6 | 75.2 | 78.4 | 36.2 | 67.1 |
| GPTSWARM (ZHUGE ET AL., 2024) | GPT-4O-MINI | 89.1 | 55.6 | 75.2 | 78.4 | 36.3 | 66.9 |
| DYLAN (LIU ET AL., 2023) | GPT-4O-MINI | 90.0 | 59.2 | 81.2 | 80.0 | 40.9 | 70.3 |
| MAS-GPT (YE ET AL., 2025) | GPT-4O-MINI | 90.2 | 61.5 | 81.2 | 80.4 | 42.6 | 71.2 |
| MAS-GPT (YE ET AL., 2025) | QWEN3-4B | 91.5 | 39.9 | 89.2 | 80.5 | 63.6 | 72.9 |
| **OURS** | GPT-4O-MINI | 92.3 | **64.7** | 83.3 | **82.0** | 44.1 | 73.3 |
| **OURS** | QWEN3-4B | **94.4** | 64.0 | **89.5** | 81.7 | **63.8** | **78.7** |

*Table 2.* Ablation study on training stages across different model scales. Qwen3-4B and Qwen3-30B denote Qwen3-4B-Instruct-2507 and Qwen3-30B-A3B-Instruct-2507, respectively. w/ SFT and w/ RL denote the Architectural Distillation and Architectural Exploration training stages, respectively.

| DRIVING MODEL | STAGE | MATH | GPQA | HOTPOTQA |
|---|---|---|---|---|
| QWEN3-4B | VANILLA | 82.1 | 55.1 | 45.5 |
| | W/ SFT | 83.3$_{(+1.2)}$ | 56.1$_{(+1.0)}$ | 47.9$_{(+2.4)}$ |
| | W/ RL | 89.5$_{(+7.4)}$ | 64.1$_{(+9.0)}$ | 61.5$_{(+16.0)}$ |
| QWEN3-30B | VANILLA | 88.7 | 70.2 | 65.5 |
| | W/ SFT | 89.8$_{(+1.1)}$ | 71.7$_{(+1.5)}$ | 67.2$_{(+1.7)}$ |
| | W/ RL | 92.5$_{(+3.8)}$ | 73.2$_{(+3.0)}$ | 70.8$_{(+5.3)}$ |

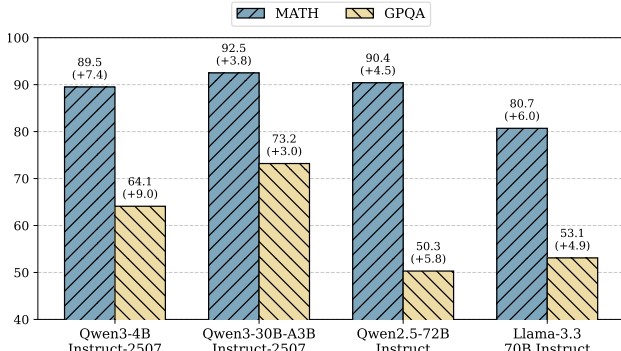

*Figure 4.* Performance comparison of MAS generated by MAS-Architect driven by different models on MATH and GPQA benchmarks. Values atop bars indicate accuracy and relative improvement over the vanilla baseline.

a remarkable 16.0% on HotpotQA, underscoring the critical role of exploration in mastering complex multi-hop reasoning tasks. Furthermore, this scalability persists in the larger 30B model; despite starting from high-performance baselines (e.g., a Vanilla score of 88.7 on MATH), the RL stage continues to push the upper bounds of capability with consistent gains between 3.0% and 5.3%, strongly validating the robustness and generalization of our progressive training strategy.

### 4.4. Efficiency Analysis

Figure 3 illustrates the efficiency analysis on the GSM8K benchmark, mapping the Pareto frontier of model performance against computational cost (measured in Tokens per Query). The visualization reveals distinct clusters of efficiency: heavy-weight baselines such as LLM-Debate exhibit the highest resource consumption, peaking around 20,000 tokens per query, while DyLAN (Liu et al., 2023) and GPTSwarm (Zhuge et al., 2024) also incur substantial costs exceeding 10,000 tokens for varying levels of accuracy. In contrast, while GDesigner and MAS-GPT represent a shift

towards more efficient architectures, our MAS-Architect distinguishes itself by occupying the optimal trade-off position in the bottom-right corner. It achieves a state-of-the-art accuracy of 94.4% with a minimal inference burden of only 2, 533 Tokens/Query, demonstrating that our framework delivers superior reasoning performance while requiring significantly fewer computational resources than existing Auto-MAS solutions. Following the G-Designer (Zhang et al., 2024a) setup, Figure 3 reports input-token cost under our Qwen3-4B-Instruct-2507 reimplementation; Appendix A further reports total-token cost, including both prompt and completion tokens.

### 4.5. Cross-model Generalizability

Figure 4 presents a detailed quantitative analysis of the MAS-Architect framework's performance across four distinct large language models on the MATH and GPQA bench-

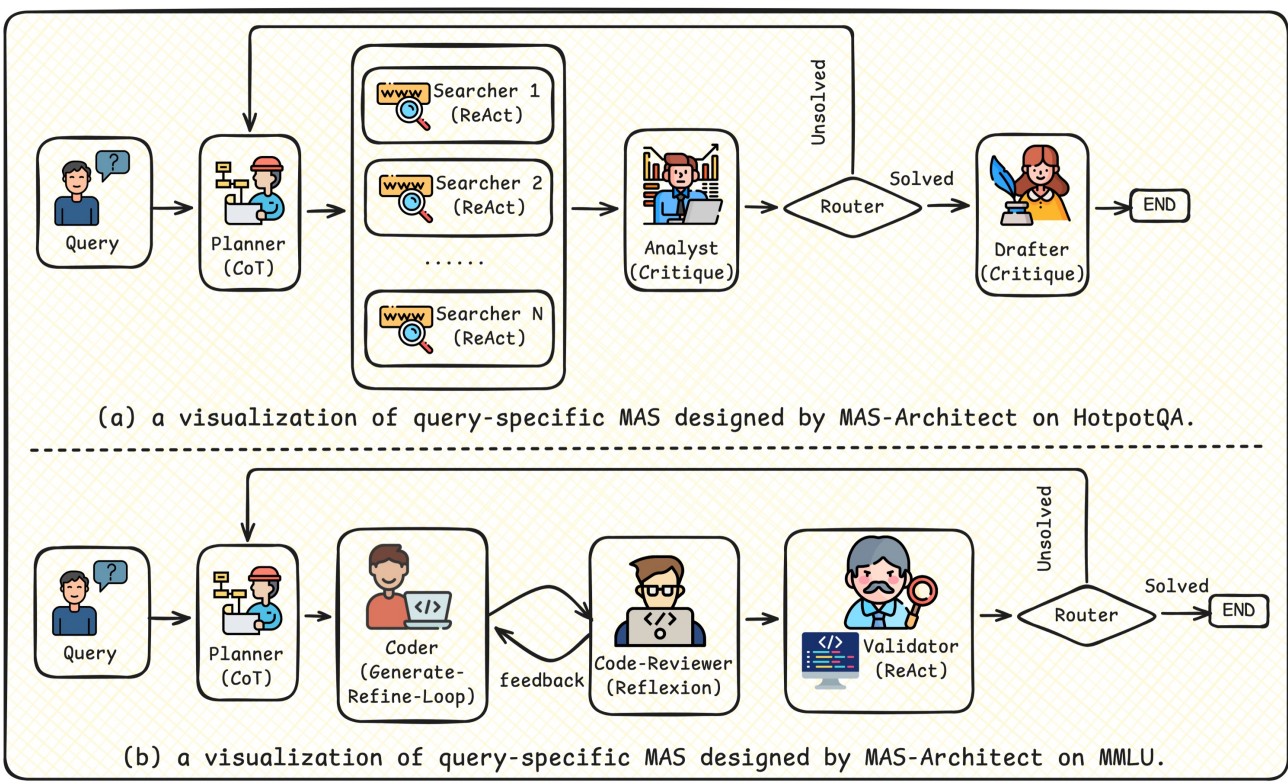

*Figure 5.* Visualization of query-specific MAS architectures synthesized by MAS-Architect. (a) On HotpotQA, the system exhibits Parallelized Reasoning-Exploration Streams using parallel ReAct-based searchers for robust retrieval. (b) On MMLU, it adopts Hierarchical Dual-Loop Refinement, featuring inner-loop code correction and outer-loop execution validation.

marks, highlighting both the resulting accuracy scores and the absolute improvements (indicated in parentheses) over vanilla baselines. For the Qwen3-4B Instruct-2507 model, the framework achieves an accuracy of 89.5% on MATH with a significant gain of +7.4%, and 64.1% on GPQA with a marked improvement of +9.0%. The Qwen3-30B-A3B Instruct-2507 model demonstrates the highest absolute peaks in this comparison, securing a MATH accuracy of 92.5% (+3.8%) and a GPQA accuracy of 73.2% (+3.0%). The Qwen2.5-72B Instruct model follows with strong results, recording 90.4% accuracy on MATH (+4.5% improvement) and 50.3% on GPQA (+5.8% improvement). Finally, demonstrating cross-architecture effectiveness, the Llama-3.3 70B Instruct model achieves an accuracy of 80.7% on MATH (+6.0%) and 53.1% on GPQA (+4.9%), consistently validating the framework's ability to drive performance enhancements across diverse model families and scales.

### 4.6. Case Study

Figure 5 illustrates the query-specific MAS architectures synthesized by MAS-Architect. In Figure 5(a), for the search-intensive HotpotQA, we observe an emergent topology characterized as **Parallelized Reasoning-Exploration**

**Streams**. By instantiating multiple ReAct-based Searchers in parallel, this structure effectively balances the depth of chain-of-thought reasoning with the breadth of extensive exploration, significantly enhancing retrieval robustness; specific implementation details are provided in Appendix E.1. In contrast, Figure 5(b) introduces **Hierarchical Dual-Loop Refinement** for MMLU. In this design, the inner loop (Coder-Reviewer) facilitates rapid code refinement to prevent expensive overhead for trivial syntax corrections, while the outer loop, governed by the Validator, leverages execution feedback to guide the Planner, enabling the system to strategically re-plan and break out of local optima (refer to Appendix E.2 for the implementation). Both instances effectively exemplify the architectural reasoning capabilities of MAS-Architect. Additionally, please refer to Appendix D for more topological patterns emerging during the reasoning process of MAS-Architect.

## 5. Conclusion

We present MAS-Architect, a framework that automates Multi-Agent System design via a novel Declarative MAS Paradigm. By decoupling topology planning from node implementation and employing a *Distill-then-Explore* training strategy, our approach enables the from-scratch gen-

eration of task-adaptive architectures. Empirical results confirm that MAS-Architect establishes a new efficiency-performance Pareto frontier, outperforming SOTA methods by 2.9% while reducing token consumption by 23.8% on GSM8K, and demonstrating robust generalization across held-out benchmarks. Furthermore, the autonomous emergence of sophisticated topological patterns, such as hierarchical refinement loops, validates the model's capacity for genuine architectural reasoning beyond simple imitation.

## Impact Statement

This paper presents work whose goal is to advance the automated design of LLM-based multi-agent systems for complex reasoning tasks. The potential positive impacts include reducing manual engineering effort, improving the accessibility of multi-agent system construction, and lowering inference token consumption for benchmark reasoning workloads. Potential risks include the misuse of automatically generated agent workflows, the propagation or amplification of model hallucinations and biases through multi-agent interactions, and unsafe behavior when tool-using agents are deployed without adequate safeguards. Our experiments are conducted on standard academic benchmarks and do not involve personal or sensitive data, nor do they target deployment in high-stakes domains such as medicine, law, or finance. We encourage practitioners to apply human oversight, safety evaluation, tool-use constraints, and domain-specific validation before deploying generated multi-agent systems in real-world settings.

## Acknowledgements

This work was supported by the National Natural Science Foundation of China under Grants 42394060 and 42394064, and by Ant Group.

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

## A. Further Comparison of Token Efficiency and Performance

This appendix reports total-token cost, defined as the sum of input (prompt) and completion tokens, complementing the input-token efficiency analysis in Figure 3. As shown in Table 3, MAS-Architect outperforms MAS-GPT in terms of both efficiency and performance. Our method significantly reduces token consumption by 8.8% to 23.8% across all benchmarks while delivering consistent performance gains. Notably, on the GSM-Hard task, the model achieves a substantial accuracy boost of 24.1 points while cutting computational overhead by 16.5%. Unlike Figure 3, which counts only prompt tokens, the statistics in this table include both prompt and completion tokens.

*Table 3.* Performance and Total Token Consumption Comparison

| Method | GSM8K | | GSM-Hard | | MMLU | | GPQA | |
|---|---|---|---|---|---|---|---|---|
| | Token Consumption | Performance | Token Consumption | Performance | Token Consumption | Performance | Token Consumption | Performance |
| MAS-GPT | 4824 | 91.5 | 8368 | 39.9 | 7171 | 80.5 | 19002 | 63.6 |
| MAS-Architect (Ours) | $3677_{(-23.8\%)}$ | $94.4_{(+2.9)}$ | $6988_{(-16.5\%)}$ | $64.0_{(+24.1)}$ | $6537_{(-8.8\%)}$ | $81.7_{(+1.2)}$ | $14633_{(-23.0\%)}$ | $63.8_{(+0.2)}$ |

## B. Fine-Grained Ablation on Topology and Node Implementation

To isolate the contribution of each design dimension in the declarative paradigm, we compare MAS-Architect with three constrained variants. "Generative" denotes free-form generation by the Meta-Agent, whereas "Fixed" denotes selection from a predefined candidate pool. Simple MAS uses a fixed star topology with one orchestrator and four workers, which slightly exceeds the average node count of MAS-Architect-generated systems. As shown in Table 4, both topology planning and node implementation provide additive gains, supporting the separation-of-concerns design.

*Table 4.* Fine-grained ablation of topology planning and node implementation.

| Setting | Topology | Node Impl. | Avg. |
|---|---|---|---|
| Full | Generative | Generative | 78.6 |
| Fixed Topology | Fixed | Generative | 76.2 |
| Fixed Nodes | Generative | Fixed | 74.7 |
| Simple MAS | Fixed | Fixed | 73.9 |

## C. Additional Training Details

We herein provide further implementation details for MAS-Architect. Specifically, for the *Architectural Distillation* stage, we employed LlamaFactory (Zheng et al., 2024) as our SFT framework; the detailed training hyperparameters are listed in Table 5. For the *Architectural Exploration* stage, we utilized verl (Sheng et al., 2024) as our RL framework, with the corresponding hyperparameters presented in Table 6.

*Table 5.* Hyperparameter Configuration for SFT

| Category | Parameter | Value | Description |
|---|---|---|---|
| *Model & Strategy* | | | |
| Model | Base Model | Qwen3-Coder-30B | Instruct Version |
| Stage | Training Stage | SFT | Supervised Fine-Tuning |
| Method | Finetuning Type | Full | Full parameter update |
| Optimization | DeepSpeed | ZeRO-3 | `ds_z3_config.json` |
| Attention | Mechanism | Flash Attention 2 | `fa2` |
| Memory | Gradient Checkpointing | True | Saves memory |
| *Training Hyperparameters* | | | |
| Batch Size | Per Device Batch | 1 | `per_device_train_batch_size` |
| Batch Size | Grad Accumulation | 4 | `gradient_accumulation_steps` |
| Precision | Mixed Precision | BF16 | Bfloat16 enabled |
| Learning Rate | Initial LR | $1.0 \times 10^{-5}$ | Peak learning rate |
| Scheduler | LR Scheduler | Cosine | Type |
| Warmup | Warmup Ratio | 0.1 | 10% of total steps |
| Duration | Num Train Epochs | 1.0 | Single pass |
| *Data Processing* | | | |
| Data | Template | qwen3_nothink | Chat template |
| Data | Cutoff Length | 8192 | Max sequence length |
| Data | Max Samples | 100,000 | Dataset cap |
| Data | Preprocessing Workers | 16 | Parallel processing |

*Table 6.* Hyperparameter Configuration for RL Training

| Category | Parameter | Value | Description |
|---|---|---|---|
| *Model & Parallelism* | | | |
| Model | Base Model | Qwen3-Coder-30B | Full Parameter SFT Init |
| Parallelism | Tensor Parallel (TP) | 4 | `tensor_model_parallel_size` |
| Parallelism | Sequence Parallel (SP) | 4 | `ulysses_sequence_parallel_size` |
| Parallelism | FSDP Size | 8 | Per node (8 GPUs) |
| *Training & Optimization* | | | |
| Batch Size | Train Prompt Batch | 32 | Global batch size |
| Batch Size | PPO Mini Batch | 8 | Update batch size |
| Batch Size | Micro Batch | 1 | Per GPU micro batch |
| Optimization | Actor LR | $1 \times 10^{-6}$ | Learning rate |
| Optimization | Critic LR | $2 \times 10^{-6}$ | Learning rate |
| Optimization | Weight Decay | 0.1 | Optimizer setting |
| Optimization | Grad Clip | 1.0 | Gradient clipping norm |
| Schedule | Warmup Steps | 10 | Linear warmup |
| Duration | Total Epochs | 50 | Training iterations |
| *Algorithm (SAPO) & PPO* | | | |
| Algo | Advantage Estimator | GRPO | Group Relative Policy Optimization |
| Algo | Loss Mode | SAPO | Shared Actor-Policy Optimization |
| Algo | $\tau_+$ (Tau Pos) | 1.0 | SAPO positive constraint |
| Algo | $\tau_-$ (Tau Neg) | 1.05 | SAPO negative constraint |
| GAE | $\gamma$ (Gamma) | 1.0 | Discount factor |
| GAE | $\lambda$ (Lambda) | 0.95 | GAE smoothing parameter |
| KL | KL Coef | 0.001 | Configured (Flag: `False`) |
| KL | KL Loss Coef | 0.001 | Configured (Flag: `False`) |
| *Tokenization & Generation* | | | |
| Length | Max Prompt Length | 2048 | Tokens |
| Length | Max Response Length | 8192 | Tokens |
| Length | Total Context | 10240 | Tokens |
| Rollout | Number of Rollouts | 8 | `actor_rollout_ref.rollout.n` |
| Sampling | Temperature | 1.0 | Generation setting |
| Sampling | Top P | 1.0 | Generation setting |
| Reward | Overlong Penalty | 1.0 | Factor for > 4096 tokens |

## D. Qualitative Analysis & Emergent Behaviors

In this section, we analyze the topological patterns autonomously emerged from MAS-Architect. We present the visualization of the generated graphs alongside their core logic implementation, demonstrating how the model adapts architectures to task complexity.

### D.1. Emergent Pattern I: Recursive Audit Chains

For hallucination-prone tasks (e.g., complex movie queries in HotpotQA), the model spontaneously generates a closed-loop structure containing a **Critic** and a **Router**.

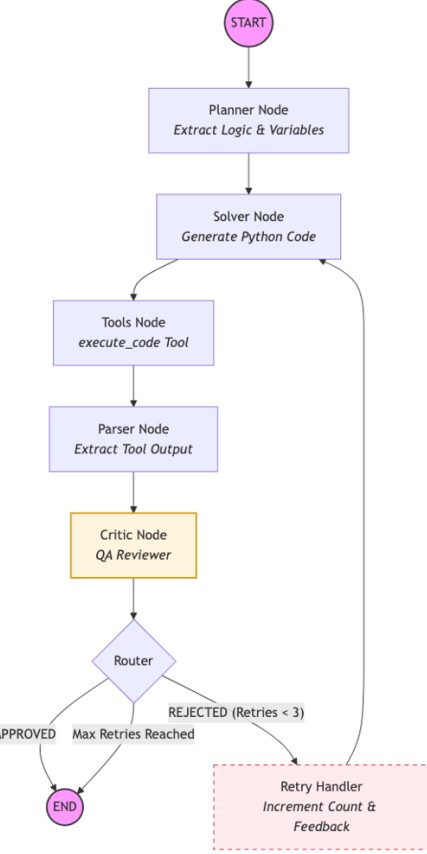

Figure 6. **Recursive Audit Chain.** The model introduces a feedback loop where the `Critic` node reviews the `Solver`'s output before final submission. The figure illustrates the workflow generated for the HotpotQA query: "Who is the director of the 2003 film which has scenes in it filmed at the Quality Cafe in Los Angeles?"

**Core Logic Implementation:** The generated code implements a conditional edge based on the Critic's feedback, enabling self-correction.

```python
# Generated Snippet: Critic & Routing Logic
async def critic_node(state: State):
    # ... (Omitted: Call LLM to evaluate answer quality) ...
    return {"status": "approved" if score > 0.8 else "rejected"}

def route_logic(state: State):
    if state["status"] == "approved":
        return END
    if state["retries"] >= 3:
        return END
    return "retry_handler"  # Backtrack loop

# Graph Definition
workflow.add_conditional_edges("critic_node", route_logic)
```

Figure 7. **Logic Implementation for Recursive Audit.** The code snippet demonstrates the conditional routing logic where the system checks the critic's feedback to decide whether to loop back or output the final answer.

## D.2. Emergent Pattern II: Conditional Parallelism

For queries requiring information from independent sources (e.g., "Address of Location A and Location B"), the model employs a **Map-Reduce** strategy to minimize latency.

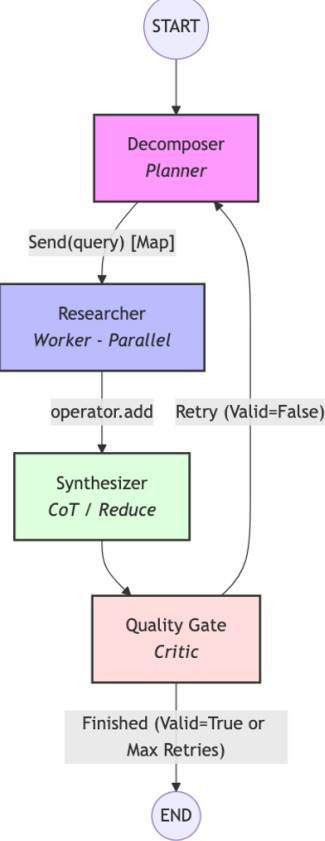

*Figure 8.* **Conditional Parallelism.** The Decomposer fans out tasks to parallel `Researcher` nodes, which are then aggregated by the `Synthesizer`. The figure illustrates the workflow generated for the HotpotQA query: "750 7th Avenue and 101 Park Avenue, are located in which city?"

**Core Logic Implementation:** The model uses `asyncio` patterns (implicit in the graph structure) to execute independent search tasks concurrently.

```
# Generated Snippet: Parallel Fan-Out
async def researcher_node(state: State):
    # ... (Omitted: Perform search based on sub-questions) ...
    return {"observations": [result]}

# Graph Definition
workflow.add_node("researcher", researcher_node)
# This Edge definition implies parallel execution, typically triggered by the upstream node's Fan-out
workflow.add_edge("decomposer", "researcher")
workflow.add_edge("researcher", "synthesizer") # Fan-in
```

*Figure 9.* **Logic Implementation for Parallelism.** The code shows the use of asynchronous execution to handle multiple research sub-tasks simultaneously, reducing overall latency.

## D.3. Emergent Pattern III: Domain-Specific Adaptation

Unlike static methods that use generic "Worker" nodes, MAS-Architect generates **Specialized Agents** tailored to the query semantics.

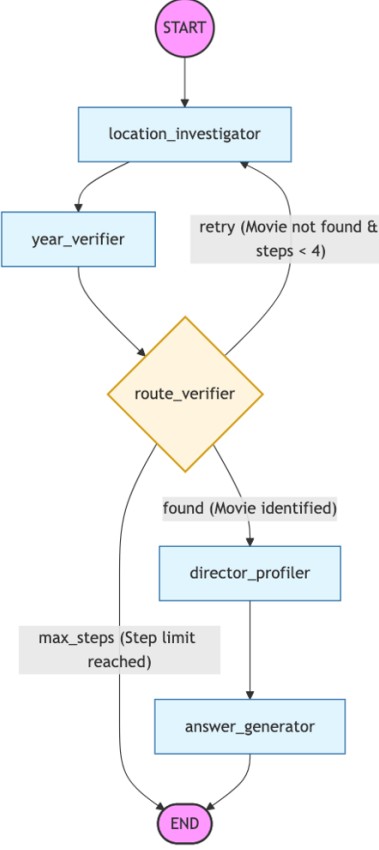

*Figure 10.* **Domain-Specific Adaptation.** The model generates role-specific agents (`Location Investigator`, `Year Verifier`) instead of generic solvers. The figure illustrates the workflow generated for the HotpotQA query: "Who is the director of the 2003 film which has scenes in it filmed at the Quality Cafe in Los Angeles?"

**Core Logic Implementation:** The model generates unique system prompts for each node, effectively "hiring" experts on the fly.

```
# Generated Snippet: Specialized System Prompts
location_agent = create_agent(
    model,
    tools=[search_tool],
    system_prompt="You are a Location Investigator. Verify filming locations..."
)

year_agent = create_agent(
    model,
    tools=[search_tool],
    system_prompt="You are a Year Verifier. Confirm the release date..."
)
```

*Figure 11.* **Logic Implementation for Agent Specialization.** The code snippet highlights how specific system prompts are dynamically assigned to different nodes based on the task requirements.

# E. Implementation Details of the Visualizations

This section provides the concrete code implementations corresponding to the architectures discussed in the main text (Figure 5). These code snippets demonstrate the exact logic synthesized by MAS-Architect for specific benchmarks.

## E.1. Parallelized Reasoning-Exploration Streams (HotpotQA)

As shown in Figure 5(a), for retrieval-heavy tasks in HotpotQA, the system employs parallel ReAct-based searchers to ensure efficient information retrieval across multiple entities.

```python
import operator
import os
import asyncio
from typing import Annotated, List, TypedDict, Literal

from langchain.chat_models import init_chat_model
from langchain_core.prompts import ChatPromptTemplate
from langchain_core.pydantic_v1 import BaseModel, Field
from langgraph.graph import StateGraph, END
from langgraph.constants import Send
from langgraph.prebuilt import create_agent
from langchain_core.tools import tool

try:
    from src.utils import wiki_search, execute_code
except ImportError:
    def wiki_search(query): return "Mock Search Result"
    def execute_code(code_str): return "Mock Code Result"

@tool
def search_tool(query: str):
    """Search Wikipedia for information."""
    return wiki_search(query=query)

@tool
def code_tool(code_str: str):
    """Execute Python code for calculation or logic."""
    return execute_code(code_str=code_str)

worker_tools = [search_tool, code_tool]

# =======================================
# 1. Global Model Initialization
# =======================================
model = init_chat_model(
    model=os.getenv("OPENAI_MODEL", "gpt-4o"),
    base_url=os.getenv("OPENAI_BASE_URL"),
    api_key=os.getenv("OPENAI_API_KEY"),
    model_provider="openai",
    temperature=0.1
)

# =======================================
# 2. State Definitions
# =======================================
class GlobalState(TypedDict):
    original_query: str
    aggregated_context: str
    current_plan: List[str]
    search_outputs: Annotated[List[str], operator.add]
    status: Literal["SOLVED", "UNSOLVED"]
    final_answer: str
    steps_count: int

class WorkerState(TypedDict):
    search_query: str

class CoTPlan(BaseModel):
    reasoning_trace: str = Field(description="Step-by-step thinking process.")
    queries: List[str] = Field(description="List of specific search queries.")

class AnalystCritique(BaseModel):
    summary: str = Field(description="Consolidated facts.")
    missing_info: str = Field(description="What information is still missing?")
    status: Literal["SOLVED", "UNSOLVED"] = Field(description="Decision.")

# =======================================
# 3. Node Functions
# =======================================
async def planner_node(state: GlobalState):
    query = state["original_query"]
    context = state.get("aggregated_context", "None")

    prompt = ChatPromptTemplate.from_messages([
        ("system", "You are the Chief Planner. Use Chain of Thought reasoning.
```

*Figure 12.* **Implementation of Parallel ReAct (Figure 5a).** The code demonstrates how the Decomposer splits the query and aggregates results from parallel searchers (Part 1).

```
                Output the reasoning and the list of queries."),
            ("user", "User Query: {query}\n\nContext So Far:\n{context}")
        ])

        planner = prompt | model.with_structured_output(CoTPlan)
        response = await planner.ainvoke({"query": query, "context": context})

        return {
            "current_plan": response.queries,
            "search_outputs": [],
            "steps_count": state.get("steps_count", 0) + 1
        }
async def searcher_node(state: WorkerState):
    query = state["search_query"]

    # create a cot agnet
    agent = create_agent(model, worker_tools)

    try:
        result = await agent.ainvoke({"messages": [("user", f"Find detailed information
            about: {query}")]})
        output_content = result["messages"][-1].content
    except Exception as e:
        output_content = f"Search failed: {str(e)}"

    return {"search_outputs": [f"Query: {query}\nResult: {output_content}"]}

async def analyst_node(state: GlobalState):
    results = state["search_outputs"]
    current_context = state.get("aggregated_context", "")
    query = state["original_query"]

    prompt = ChatPromptTemplate.from_messages([
        ("system", "You are the Lead Analyst. Synthesize facts and critique if the query
            is answered."),
        ("user", "Original Query: {query}\n\nOld Context: {context}\n\nNew Search Results:
            \n{results}")
    ])

    analyst = prompt | model.with_structured_output(AnalystCritique)
    response = await analyst.ainvoke({"query": query, "context": current_context, "results":
        results})

    new_context = f"{response.summary}\n(Missing: {response.missing_info})"

    return {
        "aggregated_context": new_context,
        "status": response.status
    }
async def drafter_node(state: GlobalState):
    context = state["aggregated_context"]
    query = state["original_query"]

    prompt = ChatPromptTemplate.from_messages([
        ("system", "You are a professional writer. Answer the user query based ONLY on the
            provided context."),
        ("user", "Query: {query}\n\nConfirmed Facts: {context}")
    ])

    chain = prompt | model
    res = await chain.ainvoke({"query": query, "context": context})

    return {"final_answer": res.content}

# ==========================================
# 4. Routing & Graph Construction
# ==========================================

def map_to_searchers(state: GlobalState):
    plan = state["current_plan"]
    if not plan: return []
    return [Send("searcher", {"search_query": q}) for q in plan]

def router_logic(state: GlobalState):
    if state["status"] == "SOLVED": return "drafter"
    if state["steps_count"] > 3: return "drafter"
    return "planner"

workflow = StateGraph(GlobalState)

workflow.add_node("planner", planner_node)
workflow.add_node("searcher", searcher_node)
workflow.add_node("analyst", analyst_node)
workflow.add_node("drafter", drafter_node)

workflow.set_entry_point("planner")

workflow.add_conditional_edges("planner", map_to_searchers, ["searcher"])
workflow.add_edge("searcher", "analyst")
workflow.add_conditional_edges("analyst", router_logic, {"planner": "planner", "drafter": "drafter"})
workflow.add_edge("drafter", END)

app = workflow.compile()
```

*Figure 12.* **Implementation of Parallel ReAct (Figure 5a).** The code demonstrates how the Decomposer splits the query and aggregates results from parallel searchers (Part 2).

```
# ==========================================
# 5. Execution Entry
# ==========================================

if __name__ == "__main__":
    async def main():
        inputs = {
            "original_query": "In what year was the creator of the current arrangement of the
                'Simpson's Theme' born?"
        }

        final_state = await app.ainvoke(inputs)
        print(f"final_answer: {final_state['final_answer']}")

    asyncio.run(main())
```

*Figure 12.* **Implementation of Parallel ReAct (Figure 5a).** The code demonstrates how the Decomposer splits the query and aggregates results from parallel searchers (Part 3).

```
●  ●  ●                                               Untitled-1

## Architecture Blueprint

### 1. State Design

The system utilizes a dual-state architecture to manage global knowledge accumulation and parallel task execution.

#### **A. Global State (`GlobalState`)**

The central memory shared across the main workflow loop:

* **`original_query` (str)**: The user's initial question.
* **`aggregated_context` (str)**: The evolving "Knowledge Base." It acts as the long-term memory, accumulating
confirmed facts from previous iterations.
* **`current_plan` (list)**: The current set of sub-queries generated by the Planner.
* **`search_outputs` (list)**: A temporary **Accumulator** (using `operator.add`) that collects results from
parallel workers.
* **`status` (enum)**: Current state flag ("SOLVED" or "UNSOLVED") determining loop termination.
* **`final_answer` (str)**: The generated response for the user.
* **`steps_count` (int)**: Safety counter to prevent infinite research loops.

#### **B. Worker State (`WorkerState`)**

Transient state used specifically for parallel execution branches:

* **`search_query` (str)**: The specific sub-problem assigned to a single worker instance.

### 2. Nodes & Inference Patterns

The system consists of four distinct nodes organized in a **Plan-Map-Reduce-Refine** topology:

#### 1. Node: Chief Planner (The Strategist)

* **Role**: Decomposer & Orchestrator.
* **Inference Pattern**: **Chain of Thought (CoT)** + **Structured Output**.
* **Task**: Analyzes the `original_query` and the current `aggregated_context`. It generates a list of targeted
search queries (`queries`) to fill information gaps.
* **Key Logic**: It does not answer the question; it breaks it down into retrieval tasks.

#### 2. Node: Search Worker (The Parallel Agent)

* **Role**: Retrieval Specialist.
* **Inference Pattern**: **ReAct (Reason + Act)**.
* **Task**: Each instance receives one `search_query` from the Planner. It utilizes `create_agent` (internal ReAct loop)
to use tools (`search_tool`, `code_tool`) to find a specific answer.
* **Mechanism**: **Dynamic Fan-out**. Multiple instances of this node run in parallel, one for each query in the plan.

#### 3. Node: Lead Analyst (The Synthesizer)

* **Role**: Aggregator & Judge.
* **Inference Pattern**: **Synthesis & Critique**.
* **Task**: This acts as the **Reducer**. It takes the list of `search_outputs` from all workers and the old `aggregated_context`.
* **Logic**:
1. Consolidates new findings into the context.
2. Identifies what is still missing (`missing_info`).
3. Determines if the `original_query` can now be fully answered (`SOLVED` vs `UNSOLVED`).

#### 4. Node: Professional Drafter (The Writer)

* **Role**: Final Output Generator.
* **Inference Pattern**: **Grounded Generation (RAG-style)**.
* **Task**: Generates the final answer based **strictly** on the `aggregated_context`. This ensures the answer is grounded in the
retrieved facts and reduces hallucination.

### 3. Topology Logic (Iterative Map-Reduce)

The graph represents a **Dynamic Cyclic Graph** with parallel branching:

1. **Planning Phase**: The `Planner` generates a list of  queries.
2. **Mapping Phase (Dynamic Fan-out)**: The system uses a conditional edge (`map_to_searchers`) to spawn  parallel `Searcher` nodes (via `Send`).
3. **Execution Phase**: All `Searcher` nodes execute simultaneously. Their results are implicitly gathered into the `search_outputs` list.
4. **Reducing Phase**: The `Analyst` node runs once all searchers complete, synthesizing the list of results into the single `aggregated_context`.
5. **Routing/Looping**:
* **If SOLVED**: Proceed to `Drafter`  →  `END`.
* **If UNSOLVED**: Loop back to `Planner` to generate a *new* plan based on the updated context.
* **Force Exit**: If `steps_count > 3`, force proceed to `Drafter` to provide the best possible answer.

### Summary of Differences from Previous Blueprint

Unlike the previous **Nested Loop** (Coder/Critic), this architecture focuses on **Breadth-First Information Retrieval**:

* **Parallelism**: It fetches multiple pieces of evidence simultaneously (Map-Reduce).
* **Incremental Context**: It builds a knowledge base layer by layer rather than fixing a single artifact (like code).
```

*Figure 13.* **Declarative Paradigm of Parallel ReAct (Figure 5a).**

## E.2. Hierarchical Dual-Loop Refinement (MMLU)

As shown in Figure 5(b), for reasoning-intensive tasks in MMLU, the system adopts a dual-loop topology featuring iterative code refinement (Reflexion) and execution-based validation.

```python
import operator
import os
import asyncio
from typing import Annotated, List, TypedDict, Union, Optional, Dict, Any
from langchain_core.messages import BaseMessage, SystemMessage, HumanMessage
from langchain.chat_models import init_chat_model
from langgraph.graph import StateGraph, END

try:
    from src.utils import execute_code
except ImportError:
    # Mock for demonstration purposes if file is missing
    def execute_code(code_str): return "CALCULATED_VALUE: 42"

# ========================================
# 1. Global Model Initialization
# ========================================
# All nodes must share the same model instance using environment variables.
model = init_chat_model(
    model=os.getenv("OPENAI_MODEL", "gpt-4o"),
    base_url=os.getenv("OPENAI_BASE_URL"),
    api_key=os.getenv("OPENAI_API_KEY"),
    model_provider="openai",
    temperature=0
)

# ========================================
# 2. Global State Schema
# ========================================
class AgentState(TypedDict):
    user_query: str
    structured_plan: str          # Physics plan text
    python_code: str              # Current version of code
    review_feedback: str          # Criticism from Reviewer
    execution_output: str         # Actual execution result
    final_answer: Optional[str]   # Final answer

    # Loop control and meta-memory
    loop_count_code: int          # Inner loop count (Code Fix)
    loop_count_plan: int          # Outer loop count (Re-plan)
    meta_memory: List[str]        # Log of historical errors to prevent repetition

# ========================================
# 3. Node Implementations
# ========================================

async def planner_node(state: AgentState):
    """
    [Node: Strategic Planner]
    Inference Pattern: Chain of Thought (CoT)
    Focus: Variable extraction, Unit analysis, Formula selection.
    """
    print(f"\n--- [Planner] Iteration {state['loop_count_plan']} ---")
    query = state["user_query"]
    memory = "\n".join(state.get("meta_memory", []))

    system_prompt = """You are a Senior Theoretical Physicist.
    Use Chain of Thought reasoning to break down the user's problem.
```

*Figure 14.* **Implementation of Dual-Loop Reflexion (Figure 5b).** The code highlights the self-correction mechanism where the model iteratively refines its answer based on execution feedback (Part 1).

```
    CRITICAL INSTRUCTIONS:
    1. Identify all Given Variables and their Units.
    2. Identify the Unknown Variable.
    3. DETECT TRAPS: Explicitly check if units need conversion (e.g., km/h to m/s).
    4. Select the Kinematic Equation.

    Output a structured natural language plan. DO NOT write Python code yet.
    """

    user_content = f"Problem: {query}"
    if state["loop_count_plan"] > 0:
        user_content += f"\n\n[Previous Failure Context]:\n{memory}\nPlease adjust your plan accordingly."

    response = await model.ainvoke([
        SystemMessage(content=system_prompt),
        HumanMessage(content=user_content)
    ])

    return {
        "structured_plan": response.content,
        "loop_count_plan": state["loop_count_plan"] # Keep count
    }
async def coder_node(state: AgentState):
    """
    [Node: Adaptive Coder]
    Inference Pattern: Generate-Refine-Loop (Self-Correction)
    Focus: Translating Plan to Python, fixing bugs based on feedback.
    """
    print(f"\n--- [Coder] Iteration {state['loop_count_code']} ---")
    plan = state["structured_plan"]
    feedback = state.get("review_feedback", "")

    system_prompt = """You are a Python Simulation Engineer.
    Your task is to write a robust Python script based on the Physicist's Plan.

    REQUIREMENTS:
    1. Define variables clearly.
    2. Perform explicit unit conversions in code if mentioned in the plan (e.g., `v_ms = v_kmh * 1000 / 3600`).
    3. Calculate the result.
    4. Print the final result in the format: "CALCULATED_VALUE: <number>"
    """

    user_prompt = f"Plan:\n{plan}"

    # If feedback exists, it indicates entering inner loop correction mode
    if feedback:
        user_prompt += f"\n\n[CRITIC FEEDBACK - FIX REQUIRED]:\n{feedback}\nPlease rewrite the code to fix
            these issues."

    response = await model.ainvoke([
        SystemMessage(content=system_prompt),
        HumanMessage(content=user_prompt)
    ])

    # Extract code block (simple Markdown handling)
    content = response.content
    if "```python" in content:
        code = content.split("```python")[1].split("```")[0].strip()
    elif "```" in content:
        code = content.split("```")[1].split("```")[0].strip()
    else:
        code = content

    return {
        "python_code": code,
        "review_feedback": "" # Reset feedback after fixing
    }
async def critic_node(state: AgentState):
    """
    [Node: Code Critic]
    Inference Pattern: Reflexion (Static Analysis)
    Focus: Inspecting code logic BEFORE execution to catch unit errors.
    """
    print(f"\n--- [Critic] Inspecting Code ---")
    plan = state["structured_plan"]
    code = state["python_code"]

    system_prompt = """You are a Senior Code Reviewer and Physics Auditor.
    Check the Python code against the Physics Plan.

    KEY CHECKS:
    1. Did the code perform Unit Conversions (e.g., 130 km/h → m/s) BEFORE calculation? This is the
        most common error.
    2. Is the formula implemented correctly?

    Output "APPROVED" if the code looks correct.
    Output "REJECT: <reason>" if there is a logical flaw.
    """

    response = await model.ainvoke([
        SystemMessage(content=system_prompt),
        HumanMessage(content=f"Plan:\n{plan}\n\nCode to Review:\n{code}")
    ])
```

*Figure 14.* **Implementation of Dual-Loop Reflexion (Figure 5b).** The code highlights the self-correction mechanism where the model iteratively refines its answer based on execution feedback (Part 2).

```python
        decision = response.content

        if "APPROVED" in decision.upper():
            return {"review_feedback": "APPROVED"}
        else:
            return {
                "review_feedback": decision,
                "loop_count_code": state["loop_count_code"] + 1
            }
async def executor_node(state: AgentState):
    """
    [Node: Tool Executor]
    Inference Pattern: Action
    Focus: Running the approved code.
    """
    print(f"\n--- [Executor] Running Code ---")
    code = state["python_code"]

    # Call the provided execute_code tool
    try:
        result = execute_code(code_str=code)
    except Exception as e:
        result = f"Execution Error: {str(e)}"

    return {"execution_output": result}
async def validator_node(state: AgentState):
    """
    [Node: Outcome Validator]
    Inference Pattern: Reasoning & Alignment
    Focus: Matching result to options and formatting output.
    """
    print(f"\n--- [Validator] Checking Result ---")
    query = state["user_query"]
    output = state["execution_output"]

    system_prompt = """You are the Final Judge.
    1. Read the User Query (with options A, B, C, D).
    2. Read the Code Execution Output.
    3. Decide if the calculated value matches any option (allow small rounding errors).

    If matches: Output the final answer starting with "FINAL ANSWER:".
    If result is nonsensical, error, or no match: Output "RETRY: <reason>".
    """

    response = await model.ainvoke([
        SystemMessage(content=system_prompt),
        HumanMessage(content=f"Query: {query}\n\nExecution Output: {output}")
    ])

    content = response.content

    if "FINAL ANSWER" in content:
        return {"final_answer": content}
    else:
        # Record error to meta-memory for Planner's next reference
        new_memory = f"Attempt {state['loop_count_plan'] + 1} failed. Output was: {output}. Judge
            remarks: {content}"
        return {
            "final_answer": None,
            "meta_memory": state.get("meta_memory", []) + [new_memory],
            "loop_count_plan": state["loop_count_plan"] + 1,
            # Reset inner loop count for next big iteration
            "loop_count_code": 0
        }

# ========================================
# 4. Topology & Router Logic
# ========================================

def route_code_review(state: AgentState):
    """Router for Inner Loop (Code Refinement)"""
    feedback = state.get("review_feedback", "")

    # If approved or infinite loop protection (over 3 times), proceed
    if "APPROVED" in feedback.upper() or state["loop_count_code"] > 2:
        return "approved"
    return "rejected"

def route_final_validation(state: AgentState):
    """Router for Outer Loop (Strategy Correction)"""
    final = state.get("final_answer")

    if final and "FINAL ANSWER" in final:
        return "success"

    # Outer loop protection
    if state["loop_count_plan"] > 2:
        return "force_end"

    return "retry"

# Graph Construction
workflow = StateGraph(AgentState)
```

*Figure 14.* **Implementation of Dual-Loop Reflexion (Figure 5b).** The code highlights the self-correction mechanism where the model iteratively refines its answer based on execution feedback (Part 3).

```python
# Add Nodes
workflow.add_node("planner", planner_node)
workflow.add_node("coder", coder_node)
workflow.add_node("critic", critic_node)
workflow.add_node("executor", executor_node)
workflow.add_node("validator", validator_node)

# Add Edges & Conditional Edges

# 1. Start → Planner
workflow.set_entry_point("planner")

# 2. Planner → Coder
workflow.add_edge("planner", "coder")

# 3. Coder → Critic
workflow.add_edge("coder", "critic")

# 4. Critic → (Router) → Coder OR Executor
workflow.add_conditional_edges(
    "critic",
    route_code_review,
    {
        "rejected": "coder",    # Inner Loop
        "approved": "executor"  # Proceed
    }
)

# 5. Executor → Validator
workflow.add_edge("executor", "validator")

# 6. Validator → (Router) → Planner OR End
workflow.add_conditional_edges(
    "validator",
    route_final_validation,
    {
        "retry": "planner",  # Outer Loop
        "success": END,
        "force_end": END
    }
)

# Compile
app = workflow.compile()

# ==========================================
# 5. Output Format & Execution Entry
# ==========================================

if __name__ == "__main__":
    async def main():
        # 1. Instantiate the graph (already done via app = workflow.compile())

        # 2. Build initial inputs based on the Query
        inputs = {
            "user_query": "A car travels at 130 km/h. How many meters does it travel in 5 seconds?",
            "loop_count_code": 0,
            "loop_count_plan": 0,
            "meta_memory": []
        }

        # 3. Asynchronous Execution
        final_state = await app.ainvoke(inputs)

        # 4. REQUIRED OUTPUT FORMAT
        # Extract the final result string from the last message or state
        ans = final_state.get('final_answer') or "No Answer generated."
        print(f"final_answer: {{{ans}}}")

    asyncio.run(main())
```

*Figure 14.* **Implementation of Dual-Loop Reflexion (Figure 5b).** The code highlights the self-correction mechanism where the model iteratively refines its answer based on execution feedback (Part 4).

```
● ● ●                                    Untitled-1

## Architecture Blueprint

### 1. Global State Design (`AgentState`)

This is a complex state object incorporating both short-term and long-term memory:

* **`user_query` (str)**: The original question.
* **`structured_plan` (dict)**: Analytical results from the Physicist (variables, formulas, unit constraints).
* **`python_code` (str)**: The code snippet to be executed.
* **`review_feedback` (str)**: Modification suggestions from the Code Critic (**Reflexion Memory**).
* **`execution_output` (str)**: Standard output or Traceback from code execution.
* **`final_answer` (str)**: The final formatted answer.
* **`meta_memory` (list)**: Metadata used to retain lessons across cycles (prevents repetitive errors).
* **`loop_count_code` (int)**: Inner loop counter (number of code repair attempts).
* **`loop_count_plan` (int)**: Outer loop counter (number of strategy reconstruction attempts).

### 2. Nodes & Inference Patterns

The system consists of five specialized nodes organized in a **Nested Loop Topology**:

#### 1. Node: Strategic Planner (The Physicist)

* **Role**: Physics problem deconstructor.
* **Inference Pattern**: **Chain of Thought (CoT)**. It does not write code; it performs pure logical reasoning.
* **Task**: Identify , , and ; explicitly note "Trap: Unit conversion needed for km/h to m/s"; and select the formula .

#### 2. Node: Adaptive Coder (The Developer)

* **Role**: Code generator and debugger.
* **Inference Pattern**: **Generate-Refine-Loop**.
* **Task**: Generates code based on the `structured_plan`. If `review_feedback` exists, it performs **Self-Correction**
based on that feedback.

#### 3. Node: Code Critic (The Reviewer)

* **Role**: Static code analyst.
* **Inference Pattern**: **Reflexion (Static Analysis)**.
* **Task**: Inspects logic **before** the code runs. Specifically checks if unit conversions (e.g., km/h to m/s) are
implemented. If a potential bug is found, it rejects the code and sends it back to the Coder.

#### 4. Node: Tool Executor (The Runner)

* **Role**: Runtime environment.
* **Inference Pattern**: **ReAct (Action only)**.
* **Task**: Pure execution; calls the `execute_code` tool and captures `stdout` and `stderr`.

#### 5. Node: Outcome Validator (The Judge)

* **Role**: Result validation and option matching.
* **Inference Pattern**: **Reasoning & Alignment**.
* **Task**: Compares the calculation results with the A/B/C/D options. If there is a significant discrepancy or a code
error, it triggers the **Outer Loop**, requiring the Planner to rethink the strategy.

### 3. Topology Logic (Nested Cyclic Graph)

This is a non-linear Directed Acyclic Graph (DAG) variant featuring two conditional routing paths:

* **Inner Loop (Code Refinement)**: `Coder`  `Critic`  (If Rejected)  `Coder`. Ensures code logic is sound before
execution.
* **Outer Loop (Strategy Correction)**: `Executor`  `Validator`  (If Answer is Wrong/Error)  `Planner`. Handles runtime
errors or fundamental errors in the physics model.
```

*Figure 15.* **Declarative Paradigm of Dual-Loop Reflexion (Figure 5b).**

# F. Prompt Templates & Implementation Details

The prompt used to guide LLMs in generating MAS following the declarative paradigm is shown in Figure 16.

```
# Role

You are a **Meta-Agent** and a **LangGraph System Architect**. Your core capability is to orchestrate multiple
    Single Agents to build an efficient, robust Multi-Agent System (MAS) based on user needs. You pursue
    architectural depth and logical completeness, always tending to construct robust MAS through specialized
    division of labor and structured collaboration (e.g., parallel, cyclic, hierarchical patterns).

# Task

You will receive a `User Query` and a list of `Available Tools`. Your task is to design and implement a
    **LangGraph**-based MAS.

You must think and output strictly following these two steps:

## Step 1: Design Architecture Blueprint

Before writing code, perform a systematic design. This section must include:

1. **Global State:** Define the Schema flowing through the Graph (e.g., `TypedDict` structure).
2. **Nodes:** List details for each Agent node: name, role, goal, assigned tools, and reasoning mode
    (ReAct, CoT, Reflexion, Reflection & Iteration, Plan and Execute, Self-Refine / Generate-Refine-
    Loop, CRITIC, or other custom modes).
3. **Topology:** Describe the control flow between nodes, including deterministic edges (Edges) and
    conditional edges (Conditional Edges/Router).

[IMPORTANT] Refuse to rely on a single Agent to complete all work. You should decompose complex tasks into
    a refined system of specialized collaboration and match the strongest reasoning mode to the node's
    function—for example, use Plan-and-Execute for Planners, Reflexion for Executors, and CRITIC mode for
    Reviewers. Architecturally, break linear constraints and actively adopt non-linear topologies: enable
    parallel structures for independent tasks (such as multi-dimensional information retrieval) and include
    self-correction loops for complex reasoning to ensure continuous optimization. In this process, you must
    explicitly define specific "intra-node" reasoning strategies (including ReAct, CoT, Reflexion, or Self-
    Refine) as well as the "inter-node" macro-topology (such as DAG, Cyclic DAG, or Star network) to build
    a logically rigorous and efficiently operating intelligent system.

## Step 2: Code Implementation

Based on the blueprint, output the complete Python code.

* Use `langgraph`, `langchain`, `langchain_core`.
* **MUST** include: State definition, Node function definitions, Router logic, Graph construction process,
    and the `.compile()` step.
* The code should be directly runnable (assuming utility functions are imported).

# Output Format

**You must strictly follow the Markdown output format below. Do not output any conversational filler or
    summaries outside of this format:**

## Architecture Blueprint

"""
[Describe the architecture blueprint in detail here]
"""

## Implementation

```python
# [Provide the complete LangGraph Python code here]
```

# Inputs

**User Query:**
{query}

**Tool Descriptions:**

{tool_desc}
```

*Figure 16.* The prompt used to generate MAS.

