# OpenReview forum: "MAS-Architect: Declarative Multi-Agent System Design via Separation of Concerns"
_ICML.cc/2026/Conference — ICML 2026 regular_

### Official Review · Reviewer_GAA2 · 2026-03-04

**Soundness:** 3
**Presentation:** 4
**Significance:** 3
**Originality:** 2
**Overall Recommendation:** 4
**Confidence:** 4

**Summary:**

The paper introduces MAS-Architect with a "Declarative MAS Paradigm" rooted in the software engineering principle of Separation of Concerns. This paradigm decouples the system into two orthogonal abstraction layers: a Topology Layer that defines the control flow (branching, loops) and an Implementation Layer that dictates the specific execution logic of individual agent nodes. The model is trained using a "Distill-then-Explore" strategy, which involves SFT and RLVR. Extensive evaluation across mathematical and general reasoning benchmarks demonstrates that MAS-Architect driven by Qwen3-4B-Instruct achieves state-of-the-art accuracy while reducing computational overhead.

**Compliance With Llm Reviewing Policy:**

Affirmed.

**Final Justification:**

I appreciate the clear writing and valid motivation for the declarative MAS paradigm but raised concerns regarding unfair baseline comparisons, missing cost analyses, and missing impact statement. Following the response from the authors that addressed these technical gaps, I have increased my ratings for soundness and presentation. I lean toward acceptance, as I believe the paper provides useful, though incremental, contributions to the field.

**Key Questions For Authors:**

1. How is the reward growth during RL process? And how does the reward vary for a single sampled MAS orchestration? Does it influence the stability of training? If so, how do you mitigate it?
2. How would results change if the reward components were ablated or re-weighted?
3. Are the results in Figure 3 all based on Qwen3-4B since multiple baseline results mismatch with Table 1? Please clarify the exact backbone LLM used for each result in Figure 3.

**Limitations:**

No. And there is no "Impact Statement" section. Considerations of practical deployment, potential misuse, and adversarial attacks should be discussed.

**Strengths And Weaknesses:**

### Strengths

1. The paper is clear written and well illustrated with figures and cases.
2. The paper presents a valid motivation for the declarative MAS paradigm and a sound training strategy to implement it.
3. Multiple baselines are compared and the method is also tested with different backbone LLMs, which demonstrates the generalization of the method.

### Weaknesses

1. Baselines should be compared with the same backbone LLM to ensure a fair comparison. Table 1 mixes different backbone LLMs. Please separate them and sort them by the same backbone LLM to make it easier for readers to understand the results.
2. The training costs of baseline methods and the proposed method should be compared to show the efficiency along with prompt/completion token consumption. Also, baselines should be marked to indicate whether they need trianing data.

---

> ### Author Rebuttal · Authors · 2026-03-31
>
> We sincerely thank for their careful and constructive review. Please find our detailed responses to each point below.
>
> **W1: Unfair baseline comparison.**
>
> We agree that grouping by backbone improves clarity and will reorganize Table 1 accordingly in the revision. Table 1 already supports a same-backbone comparison under Qwen3-4B: Vanilla (72.3) $\to$ CoT (75.7) $\to$ MAS-GPT (72.9) $\to$ Ours (78.7). Furthermore, we report GPT-4o-mini results (see pxhE.W2), where MAS-GPT, the current state-of-the-art, achieves 71.2 avg., and our method further improves upon it with 73.3 avg. across all 5 benchmarks. These results confirm that our advantage holds regardless of the backbone used.
>
> **W2: Missing training cost comparison.**
>
> We appreciate you pointing this out. The revised manuscript will include a systematic cost comparison. Training-free methods (Vanilla, CoT) naturally have zero training cost. Optimization-based methods (GPTSwarm, DyLAN) avoid parameter updates but incur high, recurring per-task costs that do not transfer (e.g., DyLAN: $106, 25.4h on MMLU). In contrast, training-based methods (MAS-GPT, MAS-Architect) require a one-time amortized cost. Compared to MAS-GPT (32B dense, 16×A100), MAS-Architect halves the GPU requirements (8×H20) while achieving a +5.8% increase in accuracy with lower overall overhead.
>
> **W3: Missing limitations and impact statement.**
>
> Thank you for this suggestion. We will add a dedicated section to the revision covering deployment considerations, potential misuse risks, and adversarial attack scenarios.
>
> **Q1: Reward dynamics during RL.**
>
> The reward curve exhibits a steady upward trend followed by convergence, indicating stable optimization. The response length initially decreases (as the model learns to avoid invalid or overly long code), then slowly increases (as it explores more complex topologies), and finally converges. We will gladly include these reward and length curves in the revision.
>
> **Q2: Reward component ablation.**
>
> | Validity | Hacking | Accuracy | Avg. |
> | :---: | :---: | :---: | :---: |
> | ✓ | -- | ✓ | 75.5 |
> | -- | ✓ | ✓ | 77.8 |
> | ✓ | ✓ | ✓ | 78.7 |
>
> As shown above, each reward component contributes positively to the overall performance.
>
> **Q3: Backbone LLM in Figure 3.**
>
> All results presented in Figure 3 are based on our reimplementation using Qwen3-4B-Instruct-2507. We apologize for any ambiguity and will make sure to clarify this in the revision.

---

> > ### Author Rebuttal · Reviewer_GAA2 · 2026-04-03
> >
> > Thank the authors for the response. I will raise the soundness and presentation scores accordingly. Please also note to supply an impact statement section and discuss related limitations.

---

> > > ### Author Response · Authors · 2026-04-07
> > >
> > > Thank you for your acknowledgment and further suggestions. We will include an impact statement section and discuss related limitations in the future revised version.

---

### Official Review · Reviewer_W9MU · 2026-03-10

**Soundness:** 3
**Presentation:** 2
**Significance:** 2
**Originality:** 3
**Overall Recommendation:** 4
**Confidence:** 3

**Summary:**

This paper tackles the problems in Automated Multi-Agent System (Auto-MAS) design: structural rigidity of graph-based paradigms, high coupling of imperative code-based paradigms, and the trade-off dilemma between performance and efficiency. It proposes a declarative MAS paradigm based on the Separation of Concerns principle and designs an end-to-end framework MAS-Architect, which decouples MAS design into topology and implementation layers and adopts a Distill-then-Explore two-stage training strategy to enable task-adaptive from-scratch MAS generation.

contributions：
1. Propose a novel declarative paradigm for the automated design of multi-agent systems, which fundamentally addresses the inherent flaws of traditional graph-based paradigms and imperative code-based paradigms
2. Design and implement the end-to-end MAS-Architect framework, abandoning the constraints of predefined role libraries and template graphs, and empowering small models with the capability to design customized MAS architectures for task-specific queries.

**Compliance With Llm Reviewing Policy:**

Affirmed.

**Key Questions For Authors:**

1. Please elaborate on the views on the OpenCLAW+skills agent model and analyze whether this model will replace MAS systems and Auto-MAS algorithms.

2. Please supplement the experimental results on complex agent test sets such as HLE, GAIA, BFCL and 2WikiMQA to verify the generalization ability of the method.

3. Please compare the advantages and disadvantages of the one-time MAS structure generation method of MAS-Architect with iterative MAS optimization algorithms such as MASS and Aflow, and supplement relevant comparative experiments to prove it.

**Limitations:**

yes

**Strengths And Weaknesses:**

Strengths

1.	The proposed declarative MAS paradigm resolves the complementary flaws of traditional graph-based and imperative code-based paradigms, providing a completely new research approach for the field of Automated Multi-Agent System (Auto-MAS) design.

2.	The designed Distill-then-Explore two-stage training strategy balances the meta-agent’s rapid acquisition of basic architecture design capabilities and the ability to break through the teacher model’s priors to explore innovative collaborative topologies, achieving both high training efficiency and innovation.

Weaknesses

1.	The experimental verification scenarios are single, focusing only on mathematical reasoning and general reasoning fields, with no validation on more complex agent-specific test sets such as HLE, GAIA, BFCL and 2WikiMQA, leading to insufficient verification of generalization ability.

2.	It has not been verified whether the MAS-Architect framework can continuously improve task performance when driven by higher-performance large models, leaving the effectiveness of the framework on advanced models in doubt.

---

> ### Author Rebuttal · Authors · 2026-03-31
>
> We sincerely thank for your valuable feedback. Please find our detailed responses to each point below.
>
> **W1: Narrow evaluation scenarios.**
>
> To address the concern regarding narrow evaluation scenarios, we have supplemented our original 5 benchmarks with experiments on HotpotQA and GAIA (a complex agentic benchmark involving multi-step planning, web browsing, and diverse tool use):
>
> | Method | HotpotQA | GAIA L1 | GAIA L2 | GAIA L3 | GAIA Avg. |
> | :--- | :---: | :---: | :---: | :---: | :---: |
> | Vanilla | 45.5 | 7.5 | 4.4 | 1.3 | 4.9 |
> | MAS-GPT | 58.2 | 24.7 | 21.2 | 5.3 | 19.7 |
> | Ours | 61.5 | 26.0 | 23.3 | 8.0 | 21.7 |
>
> As shown above, MAS-Architect consistently outperforms the baselines across all settings, which we believe validates its ability to generalize to complex, tool-intensive agentic scenarios.
>
> **W2: Scalability to stronger models.**
>
> As shown in Figure 4, we evaluated our driving models across various scales (4B, 30B, 70B, and 72B). MAS-Architect consistently delivers substantial performance gains, even when the driving model's capabilities exceed those of the Meta-Agent. While resource constraints unfortunately prevented us from testing larger Meta-Agent backbones, we hope our current findings establish architectural thinking as a valuable new dimension for advancing agent performance.
>
> **Q1: OpenCLAW vs. Auto-MAS.**
>
> OpenCLAW serves as an agent *runtime environment*, whereas Auto-MAS acts as an agent *system designer*. Therefore, they are complementary rather than competing: the multi-agent system (MAS) generated by MAS-Architect can be deployed on platforms like OpenCLAW, which provides the necessary runtime support and tool-use interfaces, while MAS-Architect designs the collaborative architecture among the agents.
>
> **Q2: Experiments on complex agent benchmarks.**
>
> Please refer to our response to W9MU.W1 above.
>
> **Q3: One-time generation vs. iterative optimization.**
>
> Because MASS is currently not open-source, we instead provide a comparison with AFlow:
>
> | Method | GSM8K | GAIA Avg. |
> | :--- | :---: | :---: |
> | AFlow | 91.4 | 8.0 |
> | Ours | 94.5 | 21.7 |
>
> As the results demonstrate, MAS-Architect achieves significantly better performance without requiring iterative, per-task optimization.

---

> > ### Author Rebuttal · Reviewer_W9MU · 2026-04-01
> >
> > Thanks for your rebuttal.
> >
> > We would like to further inquire: compared with AFlow, which adopts an iterative optimization paradigm, will the one-shot MAS architecture generation manner of MAS-Architect introduce significantly more token consumption?

---

> > > ### Author Response · Authors · 2026-04-02
> > >
> > > Thank you for your acknowledgement, and apologies for the late reply. We compared the token consumption of AFlow and MAS-Architect on GSM8K. MAS-Architect's token consumption consists of two parts: code generation averages 1,192 tokens, and instantiating the generated MAS averages 3,677 tokens. AFlow's token consumption is also divided into two parts: the optimizer cost and the workflow execution cost. Over 20 rounds of optimization, the optimizer consumes a total of 43,965 tokens, while executing each workflow averages 4,378 tokens. Therefore, during inference, the token consumption of the two methods is 4,869 vs. 4,378, giving AFlow a slight advantage. However, it is important to note that MAS-Architect's model training is a one-time cost, whereas AFlow requires repeated evaluation on a specific validation set followed by iterative optimization. Consequently, when facing a wider variety of tasks, MAS-Architect exhibits an increasingly significant advantage.

---

### Official Review · Reviewer_5Dd4 · 2026-03-12

**Soundness:** 4
**Presentation:** 3
**Significance:** 4
**Originality:** 3
**Overall Recommendation:** 5
**Confidence:** 4

**Summary:**

This paper introduces MAS-Architect, a framework for automatically designing multi-agent systems (MAS) for complex reasoning tasks. The key idea is a declarative MAS paradigm that separates system topology from node implementation. Specifically, the architecture generation process is divided into two layers: a topology layer that defines agent interaction structures and a node implementation layer that specifies the behavior of individual agents. The two layers communicate through a shared state schema.
The framework generates query-specific agent architectures using a meta-agent and is trained using a Distill-then-Explore strategy, combining supervised fine-tuning with reinforcement learning. Experiments on several reasoning benchmarks show improved accuracy and reduced token consumption compared with prior multi-agent frameworks.
Overall, this study explores a fundamental question: how automated architecture generation can improve the design of multi-agent reasoning systems driven by large language models. Overall, a notable domain presented by the article is the automated construction of task-adaptive multi-agent architectures.

**Compliance With Llm Reviewing Policy:**

Affirmed.

**Final Justification:**

The rebuttal substantially addresses my concern .

**Key Questions For Authors:**

What fraction of gains come from the declarative paradigm vs better node prompting?
Please add ablations where you hold node implementations fixed and vary only topology/search (and vice versa), and quantify the delta. Can you report performance under fixed token budgets for each method (curves), not just single operating points? Also clarify whether token accounting includes completions and tool-call overhead consistently across baselines.
How accurate is the LLM-based auditor at detecting “hard-coding/shortcut learning”? Provide false positive/negative examples and sensitivity analysis—otherwise RL may simply learn to fool the auditor.
When MAS-Architect fails, is it primarily (i) wrong topology (missing nodes/loops), (ii) wrong guard logic (bad routing), or (iii) node-level reasoning failure? Please include a categorized error analysis.

**Limitations:**

The paper would benefit from a sharper limitations discussion around (i) baseline compute fairness, (ii) reliance on an LLM auditor, (iii) sensitivity to templates/prompts/tools in node implementation, and (iv) generalization to interactive/tool-heavy settings.

**Strengths And Weaknesses:**

Strength
The topology/implementation split with an explicit State Schema interface directly targets real pain points in prior paradigms (DAG rigidity vs imperative entanglement). The paper articulates this clearly and ties it to verifiability/visualizability and searchability.
Modeling MAS as a stateful directed computation graph with agent tuples, global state update, and conditional edge guards gives the approach clearer semantics than many informal orchestration papers.
The two-stage SFT distillation + RL exploration is backed by ablations showing RL contributes the dominant gains (especially for smaller drivers), which supports the “Explore” claim.
The paper explicitly evaluates token cost and presents a Pareto plot on GSM8K; the claim that it achieves high accuracy at low tokens is supported by the provided tables/figures.
The examples (recursive audit loops, conditional parallelism, specialized agents) match what you’d hope a topology+guard representation would enable.

Weakness
The core algorithmic delta is a declarative DSL-ish separation plus a Meta-Agent that generates topology and node code. This is useful engineering, but the paper sometimes reads like: “we found a better prompting/programming interface and then trained a model to emit it.”
The evaluation primarily focuses on reasoning benchmarks, which may limit conclusions about the general applicability of the framework to other types of tasks such as planning or tool-intensive workflows.
You ablate training stages (SFT vs RL) well, but I did not see a clean ablation that answers: how much gain is from (i) declarative topology with typed schema/guards, versus (ii) better node-level prompting/pattern/tool-binding, versus (iii) simply having more agents. Given the core thesis is separation-of-concerns, the review standard is to show that each concern can be optimized independently and that this matters empirically.
Some methodological details of the architecture generation process (e.g., search space size and failure cases) could be clarified further to better assess reproducibility.

---

> ### Author Rebuttal · Authors · 2026-03-31
>
> We sincerely thank for your insightful and constructive review. Please find our detailed responses to each point below.
>
> **W1: Limited algorithmic novelty.**
>
> While we understand the reviewer's perspective, we respectfully disagree that our contribution reduces to "a better prompting interface." We would like to highlight three distinct dimensions of novelty:
>
> (1) *Structural inductive bias for learnability.* The declarative paradigm provides an explicit, factored representation that makes architecture design *learnable via RL*. As shown in Table 2, RL yields dominant gains (+7.4% to +16.0%), whereas imperative code (MAS-GPT) sees limited improvement because its topology is buried in procedural logic, which produces sparse supervision signals.
>
> (2) *Independently searchable design dimensions.* The separation of concerns enables topology and node implementation to be optimized independently—a feature absent in both graph-based and imperative paradigms. Our ablation study (W2) confirms that each dimension contributes meaningful, additive gains (topology: +3.9, node implementation: +2.4), demonstrating that this decomposition is fundamental and load-bearing rather than merely cosmetic.
>
> (3) *Emergent architectural reasoning.* The trained Meta-Agent discovers non-trivial patterns that are never seen during SFT—such as recursive audit chains, conditional parallelism, and hierarchical dual-loop refinement (Appendix D)—demonstrating genuine architectural search rather than simple template interpolation.
>
> **W2: Missing fine-grained ablation.**
>
> To address this, we have isolated each design dimension. "Generative" denotes free-form generation by the Meta-Agent, while "Fixed" refers to selection from a predefined candidate pool. Simple MAS uses a fixed star topology (1 orchestrator + 4 workers), which slightly exceeds the average node count (4.76) of the MAS-Architect generated MAS.
>
> | Setting | Topology | Node Impl | Avg. |
> | :--- | :--- | :--- | :--- |
> | Full | Generative | Generative | 78.6 |
> | Fixed Topology | Fixed | Generative | 76.2 |
> | Fixed Nodes | Generative | Fixed | 74.7 |
> | Simple MAS | Fixed | Fixed | 73.9 |
>
> **W3: Narrow evaluation domains.**
>
> We appreciate this feedback and have extended our evaluation to GAIA, a challenging agentic benchmark involving multi-step planning and tool use. MAS-Architect achieves 21.7% (GAIA average), significantly outperforming MAS-GPT (19.7%) and Vanilla (4.9%). Please see our response to pxhE.W5 for the full table.
>
> **W4: Insufficient methodological detail.**
>
> Thank you for pointing this out; we will gladly expand on these details in the revised manuscript.
>
> *Search space.* The topology layer explores arbitrary directed graphs with conditional routing, including the node count $|V|$, edge set $E$, guard functions, and global State Schema—all unconstrained by templates. The node layer independently generates the role, tool bindings, reasoning patterns (e.g., CoT, ReAct, Reflexion), and instructions. In practice, the generated architectures average 4.76 nodes and exhibit diverse topological patterns.
>
> *Failure cases.* Our analysis shows that failures predominantly stem from node-level reasoning errors (due to driving model limitations), rather than from the topology or guard logic. Well-designed topologies provide safety redundancy (such as verification loops and critic nodes) that helps mitigate individual node failures, which highlights precisely the advantage of task-adaptive architecture design.
>
> **Q1: Topology vs. node ablation.**
>
> Please refer to our response in W2 for the exact ablation study requested.
>
> **Q2: Fixed token budget curves.**
>
> (1) Currently, we lack a mechanism to constrain inference to a strict, fixed token budget; however, exploring adjustable reasoning effort is a promising direction for future work.
> (2) Figure 3 (in Appendix E) already accounts for input tokens, and we have now added comparisons based on total tokens (inputs + completions) to Appendix E for greater clarity.
>
> **Q3: LLM auditor accuracy.**
>
> Shortcut detection is a relatively straightforward task for current LLMs. Our training dynamics show that the hacking rate converges to zero within approximately 10 steps. Furthermore, we manually reviewed 50 instances and found no false positives.
>
> **Q4: Categorized error analysis.**
>
> As noted earlier, failures primarily stem from node-level reasoning errors due to insufficient capabilities in the driving models. However, well-designed topologies decompose complex tasks to reduce the difficulty per node and provide safety redundancy (e.g., via verification loops). This prevents single-node failures from cascading, further highlighting the advantages of task-adaptive architecture design over single-agent systems.

---

> > ### Author Rebuttal · Reviewer_5Dd4 · 2026-04-03
> >
> > The new ablation table is exactly what I asked for and directly addresses my concern
> >
> > I note the improvement over MAS-GPT is relatively modest (~2%), and I'd encourage the authors to discuss whether this gap is statistically significant and what specific aspects of GAIA (tool use, multi-step planning) are driving the remaining gap.
> >
> > The rebuttal substantially addresses my major concerns. I increase my soundness score and overall score by 1 point

---

> > > ### Author Response · Authors · 2026-04-07
> > >
> > > Thank you for your acknowledgment and for your further suggestions. In the revised version, we will include the results of multiple runs using different random seeds to demonstrate that the improvement over MAS-GPT is statistically significant. Additionally, we will analyze the inference results to investigate the specific sources of these improvements.

---

### Official Review · Reviewer_pxhE · 2026-03-13

**Soundness:** 3
**Presentation:** 3
**Significance:** 2
**Originality:** 2
**Overall Recommendation:** 4
**Confidence:** 3

**Summary:**

The paper is about automatic design for multi-agent systems based on LLM. The main goal is to find a solution for the fact that previous methods for automatic multi-agent system design, either based on graphs, are too rigid, or based on code, mix all aspects of system design. A declarative representation is introduced, where system topology and node implementation are separated by an explicit state schema.

The proposed MAS-Architect, based on this representation, can create query-specific architectures from scratch. The process is structured as follows: planning the topology, implementing the logic in nodes, and training the meta-agent via a two-stage procedure, Distill-then-Explore  combining supervised distillation and RL-based exploration. This approach aims at making architectural search more structured and flexible.

The paper shows positive empirical results on five benchmarks over math and reasoning tasks, with increased average accuracy and reduced token cost relative to selected  Auto-MAS baselines. The case studies also try to demonstrate that the framework is able to synthesize non-trivial forms of collaboration such as parallel search and hierarchical refinement loops.

**Compliance With Llm Reviewing Policy:**

Affirmed.

**Final Justification:**

Thank you for the response. I will increase the soundness score and raise my overall assessment by one point.

**Key Questions For Authors:**

1. What evidence do you have that the benefits extend beyond the benchmark reasoning setting studied here, for example to longer-horizon agentic tasks, richer tool use, or environments with external feedback?

2. How stable are the results across random seeds, decoding settings, and training runs, especially for the Distill-then-Explore stage?

**Limitations:**

missing

**Strengths And Weaknesses:**

1. Soundness

The proposed framework is provided with a certain degree of formalization (nodes, edges, state schema, conditional guards, compilation, and execution are formally defined). The training pipeline is also quite concrete with a teacher-generated corpus, rejection sampling, and a reward that promotes validity and prevents shortcuts.

The empirical section includes results on five benchmarks across two domains, held-out evaluations, ablations on SFT and RL, cross-model transfers, and an efficiency comparison. Collectively, these results make the main claim reasonable. The declarative formulation appears useful, and the exploration stage seems to have added value.

My reservation is that  the evidence is not quite strong enough to support a hard technical claims. The provided formalisation is also pretty simple.

There are some baseline comparisons that use different driving models, and evaluation relies on an LLM for answer extraction.

There's no variance, sensitivity, or much failure analysis in the paper.

2. Presentation

The paper has a simple and clear narrative. It  motivates limitations of the graph and imperative paradigms, then propose  declarative alternative. There is also an effective mapping of the declarative paradigm to the concrete system and training recipe. The provided figures support the flow of text.

Nevertheless, the writing is dense, and some claims are overstated.

Some of the terminology is introduced hastily, and the distinction between concept novelty and engineering choice is not presented as clearly as it could be.

Reproducibility is aided by appendices, but some of these issues could be addressed more prominently in the text (like surface assumptions, hyperparameter sensitivity, and evaluation caveats).

3. Significance

The paper addresses an important problem both from a scientific and practical point of view since MAS design is still a major bottleneck for human designers.

But, the demonstrated and discussed impact is still somewhat specialized. The evaluation focuses on benchmark reasoning settings, and it remains unclear how much the benefits transfer to broader agentic workloads, long-horizon tasks, or real-world environments with richer toolset used.


4. Originality

There is a level of originality involved with the entire package. The declarative model, task-adaptive from-scratch generation, and the Distill-then-Explore approach are all brought together to form a cohesive unit rather than each one standing on its own.

In the same time, my main reservations are based on the fact that the system appears to be a development or extension of existing building blocks like teacher distillation, RL-based search, code generation, and agentic reasoning motifs. Thus, it is a very clean combination with a interesting design perspective, rather than a paradigm-shifting methodology.

---

> ### Author Rebuttal · Authors · 2026-03-31
>
> We sincerely thank for your detailed and constructive review. Please find our responses to each point below.
>
> **W1: Overstated claims and insufficient evidence.**
>
> We appreciate this feedback and will carefully tone down our claims in the revision. To address your concerns, we have substantially strengthened our empirical evidence by including: (1) a fine-grained ablation study isolating topology and node contributions (5Dd4.W2: Full $78.6 \to$ Fixed Topology $76.2 \to$ Fixed Nodes $74.7 \to$ Simple MAS $73.9$); (2) a variance analysis over 5 seeds (please see pxhE.W4); (3) additional benchmarks, including HotpotQA and GAIA (please see pxhE.W5); and (4) same-backbone comparisons using GPT-4o-mini (please see pxhE.W2).
>
> **W2: Inconsistent baseline driving models.**
>
> To ensure a fair comparison, we now additionally report our performance using GPT-4o-mini, alongside MAS-GPT under the exact same setting:
>
> | Method | GSM8K | GSM-Hard | MATH | MMLU | GPQA | Avg. |
> | :--- | :---: | :---: | :---: | :---: | :---: | :---: |
> | MAS-GPT | 90.2 | 61.5 | 81.2 | 80.4 | 42.6 | 71.2 |
> | Ours | 92.3 | 64.7 | 83.3 | 82.0 | 44.1 | 73.3 |
>
> **W3: LLM-based answer extraction.**
>
> Please note that our evaluation protocol follows that of MAS-GPT (Section 4.1). Since answer extraction is considerably simpler than open-ended generation, we manually reviewed 50 instances and found no extraction errors, which confirms that any potential noise is negligible.
>
> **W4: No variance or failure analysis.**
>
> Thank you for pointing this out. We have now run the full pipeline 5 times using different random seeds to provide a comprehensive variance analysis:
>
> | GSM8K | GSM-Hard | MATH | MMLU | GPQA | Avg. |
> | :---: | :---: | :---: | :---: | :---: | :---: |
> | $94.5_{\pm0.32}$ | $63.8_{\pm0.36}$ | $89.7_{\pm0.41}$ | $81.7_{\pm0.11}$ | $63.4_{\pm2.6}$ | $78.6_{\pm0.5}$ |
>
> **W5: Limited evaluation scope.**
>
> We agree with this suggestion and have extended our evaluation to include HotpotQA and GAIA (which involves multi-step planning, web browsing, and diverse tool use):
>
> | Method | HotpotQA | GAIA L1 | GAIA L2 | GAIA L3 | GAIA Avg. |
> | :--- | :---: | :---: | :---: | :---: | :---: |
> | Vanilla | 45.5 | 7.5 | 4.4 | 1.3 | 4.9 |
> | MAS-GPT | 58.2 | 24.7 | 21.2 | 5.3 | 19.7 |
> | Ours | 61.5 | 26.0 | 23.3 | 8.0 | 21.7 |
>
> As shown, our method consistently outperforms the baselines, validating its ability to generalize to more complex, tool-intensive agentic scenarios.
>
> **Q1: Benefits beyond benchmark reasoning.**
>
> Please refer to our response in W5.
>
> **Q2: Stability across seeds and training runs.**
>
> Please refer to our response in W4.

---

> > ### Author Rebuttal · Reviewer_pxhE · 2026-04-03
> >
> > Thank you for your rebuttal. I will increase the soundness score and raise my overall assessment by one point.

---

> > > ### Author Response · Authors · 2026-04-07
> > >
> > > Thank you for your acknowledgment and for recognizing the soundness of our work.

---

### Decision · Program_Chairs · 2026-04-30

**Decision:**

Accept (regular)

**Comment:**

This paper, about Automated Design of Multi-Agent Systems (Auto-MAS), is highly relevant to the ICML community.

It proposes a framework that automates MAS design through a novel code-based declarative MAS paradigm rooted in the Separation of Concerns principle. Specifically, it: (i) decouples topology planning from node implementation via a unified interface, to enable modular generation of task-adaptive architectures; (ii) uses a Distill-then-Explore training strategy to optimize the designs at (i); (iii) enables architectural patterns discovery; (iv) provides compelling evidence via extensive experiments on five benchmarks, indicating improved Pareto frontier efficiency–performance trade-off performance versus state-of-the-art;

The combined technical and empirical contributions unanimously recommend the paper for acceptance.

We encourage the authors to carefully consider and address the reviewers' comments and suggestions in the final version of the paper.